# Discovering Physics Laws of Dynamical Systems via Invariant Function Learning

## Abstract

We consider learning underlying laws of dynamical systems governed by ordinary differential equations (ODE). A key challenge is how to discover intrinsic dynamics across multiple environments while circumventing environment-specific mechanisms. Unlike prior work, we tackle more complex environments where changes extend beyond function coefficients to entirely different function forms. For example, we demonstrate the discovery of ideal pendulum's natural motion $\alpha^2 \sin \theta_t$ by observing pendulum dynamics in different environments, such as the damped environment $\alpha^2 \sin(\theta_t) - \rho \omega_t$ and powered environment $\alpha^2 \sin(\theta_t) + \rho \frac{\omega_t}{|\omega_t|}$. Here, we formulate this problem as an *invariant function learning* task and propose a new method, known as **D**isentanglement of **I**nvariant **F**unctions (DIF), that is grounded in causal analysis. We propose a causal graph and design an encoder-decoder hypernetwork that explicitly disentangles invariant functions from environment-specific dynamics. The discovery of invariant functions is guaranteed by our information-based principle that enforces the independence between extracted invariant functions and environments. Quantitative comparisons with meta-learning and invariant learning baselines on three ODE systems demonstrate the effectiveness and efficiency of our method. Furthermore, symbolic regression explanation results highlight the ability of our framework to uncover intrinsic laws.

## 1 Introduction

Deep neural networks (Goodfellow et al., 2016) have been widely used for predicting dynamical systems (Aussem, 1999; Singh et al., 2012; Wang et al., 2016; Lusch et al., 2018; Yeo & Melnyk, 2019; Giannakis, 2019). Numerous efforts (Kirchmeyer et al., 2022; Wang et al., 2022; Mouli et al., 2024) have focused on modeling dynamical systems by forecasting future states from past observations, often emphasizing rapid adaptation to new systems or improved model architectures. However, an important scenario in scientific discovery has been overlooked, *i.e.*, learning invariant mechanisms, which aims to identify shared motion patterns across dynamical systems observed in multiple environments. This task not only facilitates scientific equation discovery but also holds potential for advancing the understanding and extraction of physical laws from observational data—such as images and videos—where physical laws are highly entangled. As the first step in invariant function learning, this paper focuses on ordinary differential equation (ODE) systems.

The need for invariant function learning arises because data collected is often observed under varying environments and entangled with multiple factors. For instance, the oscillation of a simple pendulum (Yin et al., 2021b) is commonly influenced by air friction; a prey-predator system (Ahmad, 1993) can be affected by limited resources. These factors significantly hinder deep models from learning the true and invariant dynamics. Instead of capturing invariant dynamical patterns, deep models tend to be sensitive to trivial information and spurious correlations, leading to failures in identifying the true and isolated mechanisms. In light of this challenge, we explore an innovative setting called *invariant function learning*, which aims to extract intrinsic mechanisms from data observed in multiple environments. Unlike prior work, we aim to tackle broader and more complex environments where changes extend beyond function coefficients to entirely different function forms. For example, we target the discovery of ideal pendulum's natural motion $\alpha^2 \sin \theta_t$ by observing pendulum dynamics in different environments such as the damped environment $\alpha^2 \sin(\theta_t) - \rho \omega_t$ and powered environment $\alpha^2 \sin(\theta_t) + \rho \frac{\omega_t}{|\omega_t|}$, as shown in our motivation example 1.

Invariant function learning presents two key challenges. Firstly, invariant mechanisms are not isolated entities, and being intertwined with varying initial conditions, system parameters, and time makes

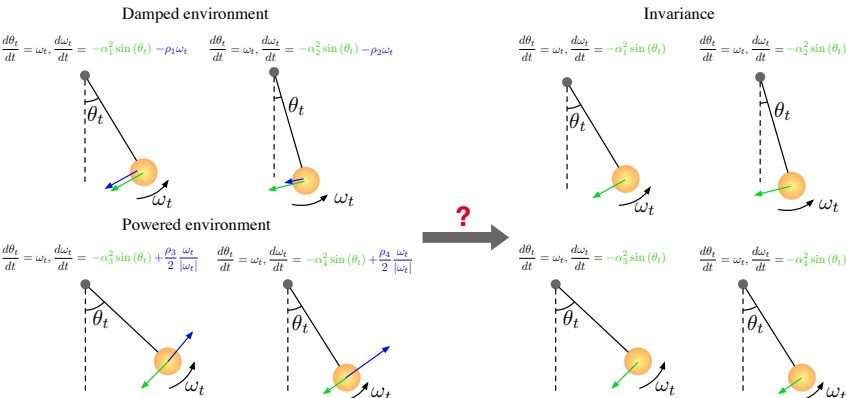

Figure 1: **Multi-environment Pendulum ODE systems.** In this example, ODEs with different coefficients and function forms are used to extract their corresponding invariant functions (green).

them difficult to define or disentangle. Secondly, existing invariant learning techniques (Arjovsky et al., 2019; Lu et al., 2021a; Rosenfeld et al., 2020; Peters et al., 2016), which are primarily designed for categorical tasks, do not extend effortlessly to dynamical systems, requiring the design of new invariant principles. To overcome these challenges, we formulate our invariant function learning framework as a causal graph where functions are parameterized and isolated to be learned and disentangled. Furthermore, we propose an invariant function learning principle and the implementation of an encoder-decoder hypernetwork (Ha et al., 2016) to identify the true invariant mechanism. Specifically, our contributions are listed as follows. (1) We introduce a new task, invariant function learning, aimed at scientific discovery. (2) We formulate its framework with causal foundations. (3) We propose an invariant function learning principle to identify true invariant mechanisms and design a method, Disentanglement of Invariant Function (DIF), with hypernetwork implementation. (4) To facilitate comprehensive benchmarking, we design several new baselines by adapting existing meta-learning and invariant learning techniques to our function learning framework and conducting extensive experimental studies.

## 2 INVARIANT FUNCTION LEARNING FOR DYNAMICAL SYSTEM

In this section, we first provide the background on ODEs, followed by the introduction and formulation of our invariant function learning task, along with the causal analyses that underpin our proposal.

### 2.1 ORDINARY DIFFERENTIAL EQUATION DYNAMICAL SYSTEM

We describe a dynamical system using an ordinary differential equation (ODE) as:

$$\frac{d\boldsymbol{x}_t}{dt} = f(\boldsymbol{x}_t), \tag{1}$$

where $\boldsymbol{x}_t \in \mathcal{X} \subseteq \mathbb{R}^d$ includes $d$ hidden states of the system at time $t$. $f \in \mathcal{F} : \mathcal{X} \mapsto T\mathcal{X}$ is the derivative function of the dynamical system mapping the hidden states to their tangent space, where $\mathcal{F}$ is the function space containing functions that describe all dynamical systems with $d$ hidden states.

Given proper time discretization, we consider $T$ time steps denoted as $t = t_0, t_1, \ldots, t_{T-1}$. The corresponding $T$-length trajectory can be written as a matrix $X = [\boldsymbol{x}_{t_0}, \boldsymbol{x}_{t_1}, \ldots, \boldsymbol{x}_{t_{T-1}}] \in \mathbb{R}^{d \times T}$. Given the system hidden states before a certain time step $T_c \in \mathbb{N}$, denoted as $X_p = X_{:,0:T_c} = [\boldsymbol{x}_{t_0}, \ldots, \boldsymbol{x}_{t_{T_c-1}}] \in \mathbb{R}^{d \times T_c}$, the forecasting task aims to predict the future trajectory $X_{:,T_c:T} = [\boldsymbol{x}_{t_{T_c}}, \ldots, \boldsymbol{x}_{t_{T-1}}]$. For theoretical analysis, we represent random variables in boldface, *e.g.*, the matrix-valued random variable corresponding to $X$ is denoted as $\mathbf{X}$; the function $f$ is a realization of the function variable f from a given function space. Full notations are detailed in Tab. 3.

### 2.2 INVARIANT FUNCTION LEARNING

In this paper, we introduce a new task, **invariant function learning (IFL)**. Specifically, given the prior distribution of trajectories, denoted as $p(\mathbf{X})$, we consider a scenario where trajectories are

Table 1: **Comparison of environments with previous works in the pendulum ODE system.** We list examples from 2 environments, the pendulum states and coefficients distribution within one environment, and inference time targets. The blue factors are changing across different environments. Full function environments are provided in Appx. C.

| Type | Environment $e = 1$ | Environment $e = 2$ | Distribution | Inference |
|---|---|---|---|---|
| Coefficient environment | $f_1 = -\alpha_1^2 \sin(\theta_t) - \rho_1 \omega_t$ | $f_2 = -\alpha_2^2 \sin(\theta_t) - \rho_2 \omega_t$ | $p(\boldsymbol{\theta}_0, \boldsymbol{\omega}_0)$ | $f_3 = -\alpha_3^2 \sin(\theta_t) - \rho_3 \omega_t$ |
| Function environment | $f_1 = -\boldsymbol{\alpha}^2 \sin(\theta_t) - \boldsymbol{\rho}\omega_t$ | $f_2 = -\boldsymbol{\alpha}^2 \sin(\theta_t) + \boldsymbol{\rho}\frac{\omega_t}{|\omega_t|}$ | $p(\boldsymbol{\theta}_0, \boldsymbol{\omega}_0, \boldsymbol{\alpha}, \boldsymbol{\rho})$ | $f_c = -\boldsymbol{\alpha}^2 \sin(\theta_t)$ |

observed under multiple environments. The trajectories observed in an environment $e \in \mathcal{E}$ are sampled from the conditional distribution $p(\mathbf{X}|e = e)$. Given a trajectory $X$ from environment $e$, our goal is to discover its invariant function $f_c$, which generates the invariant trajectory $X^c$. $f_c$ and $X^c$ only include the shared mechanisms across all environments, thus capturing the underlying natural laws unaffected by environmental factors. For instance, as illustrated in Fig. 1, in the case of a pendulum system with varying environmental effects such as frictions or power, the goal is to extract the natural motion of an ideal pendulum when excluding these external influences. A set of specific examples for the task is provided below, more examples are available in Appx. C.

**Function environments.** The environments in this paper are different from those defined in CoDA (Kirchmeyer et al., 2022), LEAD (Yin et al., 2021a), and MetaphysiCa (Mouli et al., 2024). As shown in Tab. 1, the environments/tasks used in previous works, namely, coefficient environments, are defined by the changes on the function coefficients $\alpha$ and $\rho$, *i.e.*, each environment contains only one function while different environments include functions with different coefficients. In contrast, we consider more complex cases and define environments as the interventions on function forms, *i.e.*, each environment can contain functions with the same function form and different coefficients, while different environments differ in function forms. Specifically, in Tab. 1, while coefficient environment 1 consists of a single function $f_1$, our function environment 1 includes all the functions in form of $-\boldsymbol{\alpha}^2 \sin(\theta_t) - \boldsymbol{\rho}\omega_t$ where $\boldsymbol{\alpha} \sim p(\boldsymbol{\alpha})$, $\boldsymbol{\rho} \sim p(\boldsymbol{\rho})$. Here, we model these functions as a function random variable $f_1$.

**Challenges.** However, extracting such invariant dynamics presents two significant challenges. Firstly, invariant mechanisms are intertwined with varying initial conditions, system parameters, and time, making them particularly difficult to isolate or define. Secondly, in dynamical systems, state values and their derivatives evolve over time, meaning that there is no single *invariant representation* fixed across time steps. This requires defining invariant factors in a function space, where functions can cover dynamical states. Conventional invariant learning techniques are not directly applicable in this context, as they are typically not designed to capture invariance in function spaces. These challenges necessitate the development of new function representations and the development of a novel invariant learning principle tailored to dynamical systems.

### 2.2.1 CAUSALITY-BASED DEFINITIONS

**First challenge: causal formulation.** In light of the first challenge, we aim to formulate the invariant function learning and the dynamical system forecasting problem from a causal perspective. As shown in Fig. 2, we formulate the trajectory data generation process as a Structural Causal Model (SCM) (Pearl, 2009), where c, e, and $\mathbf{X}_0$ are exogenous variables. All endogenous variables, except the function composition step $f_c, f_e \to f$, are generated with extra random noises to model complex real-world scenarios. Please refer to Appx. B.1 for more details. The optimization goal of the forecasting task is to estimate the true distribution $p(\mathbf{X})$. As can be observed from the causal graph, our function learning framework can be considered as two phases, namely, *function prediction* and *forecasting*. For the function prediction phase, similar to inverse problems (Lu et al., 2021b), given the observed trajectory $\mathbf{X}_p$, the target is to reversely infer the invariant derivative function f that can represent the dynamics of the system, *i.e.*, fitting $p(f|\mathbf{X}_p)$. Intuitively, taking Fig. 1 as an example, this phase aims at the reasoning of the function basis $\sin(\theta_t)$, $-\omega_t$, $\frac{\omega_t}{|\omega_t|}$, and the coefficients $\alpha, \rho$. After obtaining the derivative function $\hat{f}$, the forecasting phase feeds it into a numerical integrator with the initial condition $\mathbf{X}_0$ for the $\mathbf{X}$ forecasting, which can be demonstrated as $p(\mathbf{X}|\hat{f}, \mathbf{X}_0)$. Note that the bold font variables are random variables instead of individual realizations.

**Second challenge: function disentanglement.** To handle the second challenge of extracting invariant mechanisms in dynamical systems, we aim to define invariant representation in the function space, which requires the access to intermediate functions and the disentanglement formulations. With the two-phase function prediction to forecasting process, we explicitly expose the derivative function, allowing us to isolate the function prediction process and disentangle the learning of invariant functions. In the invariant function learning problem, we decompose the exposed function variable f into an invariant function variable $f_c$ and an environment-specific function variable $f_e$, as in Fig. 2. These two functions are caused by the exogenous factors c and e, respectively. Intuitively, the c variable includes the common and invariant mechanism, while e is the environment variable determined by the observational environment of the trajectory, *e.g.*, the pendulum dynamics can be observed in different mediums (environments), such as air and water.

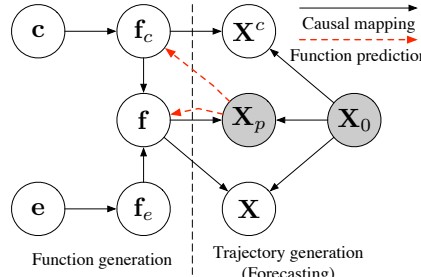

Figure 2: **Structural causal model.** The causal data generation process includes two phases: function generation and trajectory generation, which correspond to our two learning phases in parentheses, namely, function prediction and forecasting. The gray nodes in the causal graph indicate observable variables.

Formally, the invariant function learning target is the discovery of $f_c$, which eliminates the effect of environments and obtains the invariant mechanism. However, from the d-separation perspective (Pearl, 2009), since $X_p$ is the descendent of the collider f of $f_c$ and $f_e$, given $X_p$, $f_c$ and $f_e$ are correlated/biased and not distinguishable, preventing the direct fitting of $f_c$ given $X_p$. Therefore, we aim to identify this intermediate hidden variable by characterizing the unique properties of the prior distribution of $p(f_c)$. Theoretically, since f is the collider between $f_c$ and e, $f_c$ is expected to be independent of e. In addition, among all the functions that are independent of e, $f_c$ should be the most informative with respect to the observed trajectories, which will be proved in Thm. 3.1. This forms the foundation of invariant function learning.

## 3 A METHOD: DISENTANGLEMENT OF INVARIANT FUNCTION

Following the two-phase function learning framework, we now propose the first method for IFL, **D**isentanglement of **I**nvariant **F**unction (DIF), with hypernetwork-based implementations of the two corresponding networks. For the forecasting network, similar to traditional representation learning tasks, we aim to learn a function $f \in \mathcal{F} : \mathbb{R}^d \mapsto \mathbb{R}^d$. For the function prediction, however, it requires learning a function that returns a function, $h \in \mathcal{H} : \mathbb{R}^{d \times T_c} \mapsto \mathcal{F}$, *i.e.*, learning a hyper-function, which is enabled using a hypernetwork.

### 3.1 HYPERNETWORK DESIGN

**Function prediction.** To quantify the objective of the hyper-function, we approximate its output function as a neural network with $m$ parameters. The function space $\mathcal{F}$ consists of all possible neural networks with $m$ parameters, and a function $f \in \mathcal{F}$ can be represented as a vector in $\mathbb{R}^m$. Thus this parameterization process introduces a hypernetwork structure (Ha et al., 2016) into the implementation, as shown in Fig. 3. Note that since our parameterization transfers functions into the real number space, it is now possible to apply invariant learning techniques such as IRM (Arjovsky et al., 2019) and VREx (Krueger et al., 2021), where invariant (function) representations need to be extracted. In the following sections, we use $\mathcal{F}$ and omit $\mathbb{R}^m$ for simplicity.

Practically, since the number of parameters in a network is generally large, we propose to compress the invariant function representations into hidden representations, thus forming an encoder-decoder framework. Specifically, as shown in the Fig. 3, the trajectory encoder is a transformer-based network with positional embedding design, denoted as $h_{\theta_{enc}} : \mathbb{R}^{d \times T_c} \mapsto \mathcal{Z}$. Given the hidden representation from the encoder, we further encode an invariant function embedding $\hat{z}_c \in \mathcal{Z}$ and an environment function embedding $\hat{z}_e \in \mathcal{Z}$ using two multilayer perceptrons (MLPs), denoted as $h_{\theta_{inv}}$ and $h_{\theta_{env}}$, respectively. Then, aligning with our causal graph as Fig. 2, we combine $\hat{z}_c$ and $\hat{z}_e$ by summing them as the function representation $\hat{z} \in \mathcal{Z}$, which can be used for full dynamics prediction. Finally, we

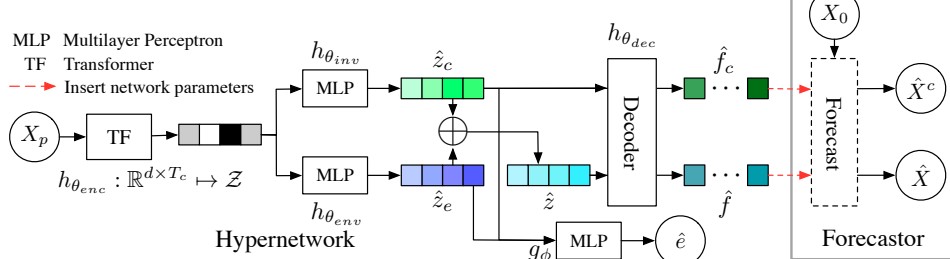

Figure 3: **DIF framework.** $\hat{e}$ denotes outputs of the discriminator $g_\phi$ introduced in Sec. 3.2.1.

learn an decoder MLP $h_{\theta_{dec}}$ to decode the function representation into $m$-dimensional neural network parameters, *i.e.*, $\hat{f}_c, \hat{f} \in \mathcal{F}$.

To facilitate theoretical analysis, we simplify the notations and denote the function prediction process in the hypernetwork as two functions $\hat{f} = h_\theta(X_p)$ and $\hat{f}_c = h_{\theta_c}(X_p)$, where $h_\theta, h_{\theta_c} : \mathbb{R}^{d \times T_c} \mapsto \mathcal{F}$; $\theta = \{\theta_{enc}, \theta_{inv}, \theta_{env}, \theta_{dec}\}$; $\theta_c = \{\theta_{enc}, \theta_{inv}, \theta_{dec}\}$. In addition, we slightly abuse the notations of $h_\theta$ and $h_{\theta_c}$ on random variables $\mathbf{X}_p$ for simplicity, producing $\hat{f} = h_\theta(\mathbf{X}_p)$ and $\hat{f}_c = h_{\theta_c}(\mathbf{X}_p)$, respectively.

**Forecasting.** Given the produced neural network function $\hat{f}$, we apply a numerical integrator as our forecastor, a function $g_{int}$ that takes a derivative function $\hat{f}$ and initial states $X_0$ as inputs, to obtain $\hat{X} = g_{int}(\hat{f}, X_0) + \epsilon$ where $\epsilon$ is sampled from a Gaussian noise $\mathcal{N}(\mathbf{X}; 0, \sigma^2 I)$ introduced by calculation deviations. This forecasting formulation enables the following probability modeling, where we obtain the forecasting given realizations $X_0$ and $\hat{f}$ as a Gaussian distribution $\mathcal{N}\left(\mathbf{X}; g_{int}(\hat{f}, X_0), \sigma^2 I\right)$ denoted as $p(\mathbf{X}|\hat{f}, X_0)$. Therefore, in probability modeling, $\hat{X}$ is sampled from $p(\mathbf{X}|\hat{f}, X_0)$. It is worth noting that unlike in inference time and analyses, it is time-consuming to use numerical integrators during training; therefore, we follow Mouli et al. (2024) and train the model by fitting the derivative $\frac{d\hat{X}_t}{dt} = \hat{f}(X_t)$ with numerical derivatives from the ground-truth $X$ instead. For simplicity, we denote $\hat{f}(\cdot)$ as a neural network based derivative function with parameters $\hat{f} \in \mathbb{R}^m$.

## 3.2 DISCOVERY OF INVARIANT FUNCTION

With the above hypernetwork design, we can now propose the discovery of the invariant function $f_c$. Following the independence and information properties of $f_c$ discussed in Sec. 2.2.1, we achieve invariant function learning with the following theorem, which is our main theoretical result.

**Theorem 3.1** (Invariant function learning principle)**.** *Given the causal graph in Fig. 2, and the predicted function random variable $\hat{f}_c = h_{\theta_c}(\mathbf{X}_p)$, it follows that the true invariant function random variable $f_c = h_{\theta_c^*}(\mathbf{X}_p)$, where $\theta_c^*$ is the solution of the following optimization process, described as*

$$\theta_c^* = \arg\max_{\theta_c} I(h_{\theta_c}(\mathbf{X}_p); f|\mathbf{X}_0) \ \ s.t. \ \ h_{\theta_c}(\mathbf{X}_p) \perp\!\!\!\perp e, \tag{2}$$

*where $I(\cdot; \cdot)$ is mutual information that measures the information overlap between the predicted invariant function random variable $h_{\theta_c}(\mathbf{X}_p)$ and the true full-dynamics function random variable $f$.*

The proof is available in Appx. B.2. Thm. 3.1 establish guarantees and conditions for the function output $\hat{f}_c$ of the hypernetwork to be the invariant function $f_c$, fulfilling the goal of the IFL task.

### 3.2.1 IMPLEMENTATION OF INVARIANT FUNCTION LEARNING PRINCIPLE

Following Thm. 3.1, next we introduce the implementation and optimization process of our proposed networks. We first train the encoder and decoder of our hypernetwork by approximating the trajectory distribution $p(\mathbf{X})$, parameterized as $p(\mathbf{X}|h_\theta(\mathbf{X}_p), \mathbf{X}_0)$, where we apply the cross-entropy minimization. Given that our supervision signals only come from the ground-truth trajectories, we introduce a simple lemma for our optimization processes. The proof is provided in Appx. B.3.

**Lemma 3.2** (ODE cross-entropy minimization)**.** *Given forecasting model $p(\mathbf{X}|h_\theta(\mathbf{X}_p), \mathbf{X}_0)$, it follows that the cross-entropy minimization between the data distribution $p(\mathbf{X})$ and $p(\mathbf{X}|h_\theta(\mathbf{X}_p), \mathbf{X}_0)$ is equivalent to minimizing mean square error $\min_\theta \mathbb{E}_{X \sim p} \|X - \hat{X}\|_2^2$, where $\hat{X}$ is sampled from $p(\mathbf{X}|h_\theta(X_p), X_0)$.*

In order to discover invariant functions, we apply the invariant function learning principle, which requires maximizing the conditional mutual information between the predicted function random variable $\hat{f}_c = h_{\theta_c}(\mathbf{X}_p)$ and f. Based on the derivation of Lemma 3.2, we have the following proposition.

**Proposition 3.3** (ODE conditional mutual information maximization)**.** *Given forecasting model* $p(\mathbf{X}|h_{\theta_c}(X_p), X_0)$*, it follows that the conditional mutual information maximization* $\max_{\theta_c} I(h_{\theta_c}(\mathbf{X}_p); \mathbf{f}|X_0)$ *is equivalent to minimizing mean square error* $\min_{\theta_c} \mathbb{E}_{\mathbf{X} \sim P} \|X - \hat{X}^c\|_2^2$*, where* $\hat{X}^c$ *is the predicted trajectory sampled from* $p(\mathbf{X}|h_{\theta_c}(X_p), X_0)$*.*

The proof is provided in Appx. B.4. Lemma 3.2 and Prop. 3.3 essentially transfers the mutual information maximization of Thm. 3.1 into an implementable optimization of mean square error (MSE) loss, enabling the practical use of the invariant function learning principle. In addition, the independence constraint in Thm. 3.1 requires that the extracted functions should not be over-informative or contain biased information from environments e. This independence constraint can be implemented in an adversarial way, where we require the environment prediction $P(e|\hat{f}_c)$ to be as less informative as possible, *i.e.*, minimizing the mutual information of $I(e; \hat{f}_c)$. Thus we introduce an environment discriminator $g_\phi$, a.k.a., $P_\phi(e|\hat{f})$, which aims to distinguish the environment of any function from $\mathcal{F}$. The hypernetwork is trained adversarially to enforce $\hat{f}_c$ as indistinguishable as possible. The theoretically analysis of this independence training is provided in Appx. B.5.

**Objective.** The overall optimization objective can be obtained with three training strategies. First, the training of the discriminator is conducted on both $\hat{f}_c$ and $\hat{f}_e$ to fully capture environment patterns. Second, we use the corresponding hidden representations of $\hat{f}_c$ and $\hat{f}_e$, $\hat{z}_c$ and $\hat{z}_e$, as the input of the discriminator. Third, as we mentioned in Sec.3.1, during training, we fit derivatives instead of using an integrator.

$$\min_\theta \mathbb{E}_{\mathbf{X} \sim p} \sum_t \left\| \frac{dX_t}{dt} - \hat{f}(X_t) \right\|_2^2 + \lambda_c \cdot \min_{\theta_c} \mathbb{E}_{\mathbf{X} \sim p} \sum_t \left\| \frac{dX_t}{dt} - \hat{f}_c(X_t) \right\|_2^2$$
$$+ \lambda_{dis} \cdot \left[ \min_\phi -\mathbb{E}_{\mathbf{X} \sim p} \log g_\phi(\hat{z}_c) + \min_{\phi, \bar{\theta}_e} -\mathbb{E}_{\mathbf{X} \sim p} \log g_\phi(\hat{z}_e) \right] + \lambda_{adv} \cdot \max_{\bar{\theta}_c} -\mathbb{E}_{\mathbf{X} \sim p} \log g_\phi(\hat{z}_c) \tag{3}$$

where we denote $\bar{\theta}_e = \{\theta_{enc}, \theta_{env}\}$; $\bar{\theta}_c = \{\theta_{enc}, \theta_{inv}\}$. Please refer to Appx. D.2.2 for more details.

**Efficient hypernetwork implementation.** Last but not least, one of the major challenges that limits the usage of hypernetworks is the implementation complexity. In this work, we propose a reference-based hypernetwork implementation to accelerate the running speed using only PyTorch Paszke et al. (2019) without re-implementing basic neural networks. The speedup compared to the naïve implementation and the vectorized functional implementation are 16.8x and 2x, respectively. Please refer to Appx. F for implementation and experimental details.

## 4 RELATED WORK

This work is inspired by the ideas and limitations of previous research in dynamical system forecasting, meta-learning, and invariant learning.

Deep learning models are widely applied in many physical applications (Lusch et al., 2018; Yeo & Melnyk, 2019; Kochkov et al., 2021; Chen et al., 2018) including partial differential equations (PDEs) with the focus on the multi-scale (Li et al., 2020; Stachenfeld et al., 2021), multi-resolution (Kochkov et al., 2021; Wu et al., 2022), and long-term stability (Li et al., 2021; Lippe et al., 2023) issues. Operator learning and neural operators (Gupta et al., 2021; Kovachki et al., 2023) are popular for PDE estimations. Although the ODE dynamical system does not contain the multi-scale problem that Fourier neural operator (Kovachki et al., 2023) tried to solve, our framework can be considered as a kind of operator learning.

Meta-learning methods (Finn et al., 2017; Rusu et al., 2018; Li et al., 2017; Zintgraf et al., 2019; Perez et al., 2018) aim to learn meta-parameters that can be used across multiple tasks, where the meta parameters are generally learned to make rapid adaptations. In previous meta-learning studies on dynamical systems (Kirchmeyer et al., 2022; Wang et al., 2022; Yin et al., 2021a), the objective was to find a meta-function that could quickly adapt to multiple new systems, where hypernetworks are only employed as low-rank adaptors for new dynamical system trajectories, similar to the idea

of LoRA (Hu et al., 2021). Our work differs from such meta-learning approaches in two key ways. First, we focus on discovering invariant functions rather than quickly adaptable ones. Second, while meta-learning methods seek to learn a single meta-function, our framework learns multiple functions, represented by an invariant function random variable $f_c$. This distinction stems from our more complex environment definition, detailed in Sec. 2.2. From another aspect, learning an invariant function distribution instead of a single function can be considered as generalized meta-learning with an invariant function learning goal.

Current invariant learning methods (Arjovsky et al., 2019; Lu et al., 2021a; Rosenfeld et al., 2020; Krueger et al., 2021; Sagawa et al., 2019) follow the framework of invariant risk minimization (IRM) (Arjovsky et al., 2019), which was inspired by invariant causal predictor (Peters et al., 2016). This invariant learning framework aims to learn a hidden invariant representation that generalizes across multiple environments, ensuring out-of-distribution performance. However, this approach cannot work on dynamical forecast tasks due to the lack of invariant function definition and the violation of the categorical data assumption. To be more specific, first, invariant functions cannot be naturally defined in the real number vector space. Second, invariant learning commonly assumes the prediction results are categorical, where a single invariant representation can fully determine the corresponding label. However, this assumption is violated in dynamical system forecasting, where the invariant mechanism is only partially responsible for the output. In this case, the IRM principle can not hold even when the invariant function ground truth is provided. To address the issues, we introduce the causal assumption (see Fig. 2) that defines the invariant function space, and propose the corresponding invariant function learning principle and implementation.

A related line of research involves symbolic regression for ordinary differential equations (ODEs), where transformer models have shown significant success (Becker et al., 2023; d'Ascoli et al., 2023; Seifner et al., 2024). While these approaches primarily focus on deriving symbolic expressions for individual trajectories, rather than identifying invariant functions across groups of trajectories, exploring the interplay between these two directions presents an exciting avenue for future research.

## 5 EXPERIMENTS

We conduct experiments to address the following research questions. **RQ1**: Are existing meta-learning and invariant learning techniques effective for extracting invariant functions? **RQ2**: Can the proposed invariant function learning principle outperform baseline techniques? **RQ3**: How do the full functions f and the invariant functions $f_c$ differ in performance? **RQ4**: Are the extracted invariant functions explainable and aligned with the true invariant mechanisms? **RQ5**: How will performance change given different lengths of inputs and types of environments? (See Appx. E) **RQ6**: Is the proposed hypernetwork implementation more efficient than previous implementations? (See Appx. F)

### 5.1 DATASETS

In our experiments, we introduce three multi-environment (ME) datasets, ME-Pendulum, ME-Lotka-Volterra, and ME-SIREpidemic. These three datasets are generated by simulators modified from the DampedPendulum (Yin et al., 2021b), Lotka-Volterra (Ahmad, 1993), and SIREpidemic (Wang et al., 2021). Specifically, each of the dataset's training sets includes four environments with 200 samples for each environment. Specifically, each environment corresponds to one specific environmental effect. ME-Pendulum contains three types of friction and one effect with external energy. ME-Lotka-Volterra modified the common predatory relationship into four modified relationships, *e.g.*, adding resource limits. ME-SIREpidemic produces four conceptual epidemiology models with the same susceptible population to infected population relationship. In addition to the training set, we generate 200 samples with 50 samples for each environment as an in-distribution test set. Please refer to Appx. C for more details.

### 5.2 EXPERIMENTAL SETUP

To quantitatively evaluate the invariant function extraction performance, we need to remove the environment-related effects to generate invariant trajectories $X^c$ as the invariant function ground-truth, *e.g.*, we simulate new data by eliminating $-\rho\omega_t$ from $-\alpha^2\sin(\theta_t) - \rho\omega_t$ in the ME-Pendulum dataset (Fig. 1). To be more specific, a generated invariant trajectory $X^c$, aligning the causal graph, has

the same system parameters as the corresponding biased trajectory $X$ for the invariant part controlled by c, *i.e.*, they have the same $\alpha$ in the pendulum example. This invariant trajectory generation is being done on the in-distribution test set so that each trajectory $X$ in this test set has its special corresponding invariant trajectory ground truth $X^c$.

This test set design enables us to mimic the situation of scientific discoveries, where we only observe environment-biased data but are required to find and evaluate invariant function candidates. Specifically, given the biased $X_p$, the hypernetwork is supposed to predict the corresponding invariant derivative function $\hat{f}_c$. Then, with a numeral integrator, the output of this invariant forecaster $\hat{X}^c$ will be evaluated by comparing with the corresponding invariant trajectory ground truth $X^c$ using normalized root mean square error (NRMSE).

### 5.3 PROPOSED META-LEARNING AND INVARIANT LEARNING BASELINES

General dynamical system forecasting is different from invariant function learning significantly, where they focus on how to adapt to new trajectories, which commonly requires further optimization, *e.g.*, test-time adaptation (Mouli et al., 2024) or adaptations with meta information (Wang et al., 2022; Kirchmeyer et al., 2022). Unfortunately, under our scientific discovery setting, there is no extra information provided at test time, making them inapplicable to this setting. Therefore, we construct 4 new baseline settings by transplanting the techniques of previous meta-learning and invariant learning to our proposed framework detailed in Appx. D.1.

We first adopt the meta-learning baseline MAML (Finn et al., 2017), where we use its learned meta-parameters for invariant learning to evaluate whether the fastest adapted parameter is the invariant function parameter. Our second meta-learning baseline is CoDA (Kirchmeyer et al., 2022), where we replace its hypernetwork decoder with the full encoder-decoder hypernetwork in our framework to fit in our task. Aligning with the original CoDA paper, we set the dimension of the hidden representation to be 2. Similar to MAML, we eliminate the adaptation part and use only the learned meta-parameter for invariant state prediction.

For invariant learning baselines, we adopted the two most typical techniques, IRM (Ahuja et al., 2021) and VREx (Krueger et al., 2021). These two techniques are applied to the proposed framework, where IRM stands for the most typical definition of invariant learning, while VREx stands for the distributionally robust optimization baseline, which can be considered as the generalization of GroupDRO (Sagawa et al., 2019).

### 5.4 QUANTITATIVE RESULTS

Similar to other scientific discovery tasks, such as drug discovery, constructing a proper validation set is challenging. Instead, with only observational data available, we generate invariant function candidates that can be further validated in real experimental settings, *e.g.*, through the introduction of interventions (Pearl, 2009).

To quantitatively compare these methods, we provide the corresponding hyper-parameter search spaces for each technique in Appx. D and plot the results of random hyper-parameter sampling as distributions using Boxen plots. As shown in Fig. 4, we compare the quality of the invariant function candidates based on their median, best result, and quantiles. Specifically, the median performance of our proposed method surpasses the middle candidates of all other approaches. The performance gaps are particularly notable on the ME-Pendulum and ME-SIR-Epidemic datasets. For example, on ME-Pendulum, over 75% of our method's candidates outperform the best results of MAML and CoDA, and more than 93.75% candidates of IRM and VREx. On ME-Lotka-Volterra, the median of our candidates still outperforms nearly all candidates from other methods. In addition, as shown in the visualizations on ME-Pendulum 5, our learned invariant function $\hat{f}_c$ eliminates environmental resistances from the original trajectory (Fig. 5a) and obtain a simple pendulum motion without attenuation (Fig. 5b). Both quantitative and visualization results demonstrate the superior capability of our method in extracting invariant functions (RQ2).

To address the first research question (RQ1), we observe that the invariant learning techniques, IRM and VREx, are generally more stable than the meta-learning baselines. Although IRM and VREx do not surpass MAML on ME-Lotka-Volterra, they outperform MAML on 2 out of 3 datasets and are consistently better than CoDA. However, when compared to our proposed method, the best function

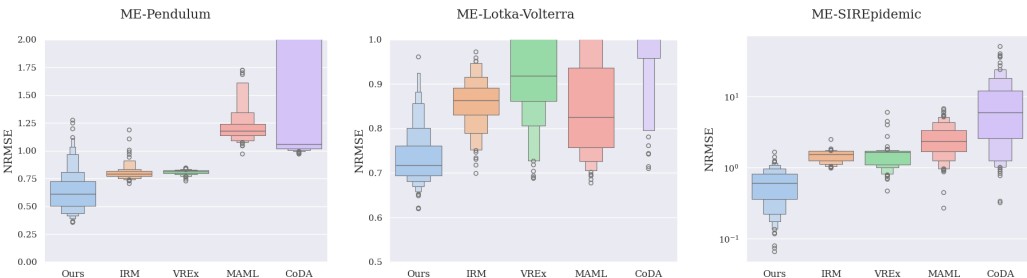

Figure 4: **Invariant trajectory prediction errors** on 5 methods under 3 multi-environment ODE systems. For each method, we provide model candidates with 80+ random hyper-parameter selections in their searching spaces, *i.e.*, more than 1200 models in the figure.

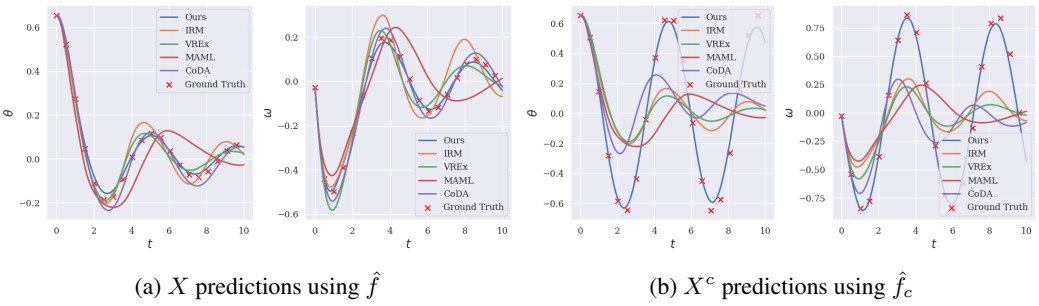

(a) $X$ predictions using $\hat{f}$          (b) $X^c$ predictions using $\hat{f}_c$

Figure 5: **Visualization of trajectory predictions on ME-Pendulum**

Table 2: **Invariant function validation and symbolic regression.** NAN denotes that the result is not applicable or not of interest.

| Target | Function | ME-Pendulum | | ME-Lotka-Volterra | | ME-SIREpidemic | |
|---|---|---|---|---|---|---|---|
| | | NRMSE | SR Explanation | NRMSE | SR Explanation | NRMSE | SR Explanation |
| $\mathbf{X}^c$ | $\hat{f}_c$ | 0.3561 | $\dfrac{d\theta_t}{dt} = 0.99\omega_t$ $\dfrac{d\omega_t}{dt} = -0.97\alpha^2 \sin(\theta_t)$ | 0.6194 | $\dfrac{dp_t}{dt} = 1.254p_t - 0.38q_tp_t$ $\dfrac{dq_t}{dt} = 4.1p_t - 0.30q_t - \gamma$ | 0.0652 | $\dfrac{dS_t}{dt} = -1.7S_tI_t$ $\dfrac{dI_t}{dt} = 0.42S_tI_t$ $\dfrac{dR_t}{dt} = -0.0088$ |
| | $\hat{f}$ | 0.7884 | $\dfrac{d\theta_t}{dt} = \omega_t \cos\left(\dfrac{\omega_t}{e^\alpha}\right)$ $\dfrac{d\omega_t}{dt} = \theta_t\alpha\left(-\alpha + \rho\right)$ | 0.7919 | $\dfrac{dp_t}{dt} = -0.76p_t$ $\dfrac{dq_t}{dt} = \dfrac{p_t}{0.36} - \gamma$ | 0.9867 | $\dfrac{dS_t}{dt} = -0.24\beta I_tS_t - 1.2$ $\dfrac{dI_t}{dt} = 0.40S_t$ $\dfrac{dR_t}{dt} = 0.66\gamma$ |
| $\mathbf{X}$ | $\hat{f}_c$ | 0.7994 | NAN | 0.6912 | NAN | 0.7641 | NAN |
| | $\hat{f}$ | 0.1700 | NAN | 0.3881 | NAN | 0.0212 | NAN |
| $f_c$ GT | | NAN | $\dfrac{d\theta_t}{dt} = \omega_t$ $\dfrac{d\omega_t}{dt} = -\alpha^2 \sin(\theta_t)$ | NAN | $\dfrac{dp_t}{dt} = \alpha p_t - \beta p_tq_t$ $\dfrac{dq_t}{dt} = \delta p_tq_t - \gamma q_t$ | NAN | $\dfrac{dS_t}{dt} = -\beta\dfrac{S_tI_t}{S_t + I_t + R_t}$ $\dfrac{dI_t}{dt} = \beta\dfrac{S_tI_t}{S_t + I_t + R_t}$ $\dfrac{dR_t}{dt} = 0$ |

candidates from these invariant learning techniques are suboptimal. This confirms that the general invariant learning principles fall short in the context of invariant function extraction, aligning with the discussions in Sec. 4.

## 5.5 FULL FUNCTION V.S. INVARIANT FUNCTION

To analyze **RQ3**, we benchmark our best invariant function learning models on the invariant state ground truth $\mathbf{X}^c$ and the multi-environment state ground truth $\mathbf{X}$, comparing their results using the predicted invariant function $\hat{f}_c$ and the full function $\hat{f}$. As shown in Tab. 2, the performance on $\mathbf{X}^c$ using $\hat{f}_c$ represents the core results of our invariant function learning approach. In contrast, the predictions

on $\mathbf{X}^c$ using $\hat{\mathbf{f}}$ serve as a baseline for the *ablation study*, where no invariant function learning principle is applied. Additionally, the prediction error on $\mathbf{X}$ using $\hat{\mathbf{f}}_c$ implies the environmental information eliminated by the invariant function learning principle, while the NRMSEs on $\mathbf{X}$ using $\hat{\mathbf{f}}$ reflect standard in-distribution (ID) test errors. We observe that the ME-Lotka-Volterra dataset is the most challenging, with an NRMSE of 0.3881 in the ID test. This result is consistent with general deep learning outcomes in Mouli et al. (2024), given that ME-Lotka-Volterra is more complex than its original version.

As expected, $\hat{\mathbf{f}}$ performs well on $\mathbf{X}$, while $\hat{\mathbf{f}}_c$ excels in predicting $\mathbf{X}^c$. In our ablation study, we compare the performance of invariant state predictions $\mathbf{X}^c$ across all datasets. The predicted invariant functions $\hat{\mathbf{f}}_c$ significantly outperform the full predicted functions $\hat{\mathbf{f}}$ in terms of NRMSE, validating the effectiveness of the proposed invariant function learning principle. Furthermore, we conduct more strict ablation study with independent $\hat{\mathbf{f}}$ and $\hat{\mathbf{f}}_c$ training in Appx. E.1.

### 5.6 Symbolic Regression Explanation

Furthermore, to address **RQ4**, we analyze the extracted invariant functions $\hat{\mathbf{f}}_c$ by applying symbolic regression using PySR (Cranmer, 2023). As shown in Tab. 2, we compare the symbolic regression (SR) explanations of the extracted invariant functions $\hat{\mathbf{f}}_c$ with the true invariant functions $\mathbf{f}_c$. On the ME-Pendulum dataset, the frictionless pendulum function is nearly perfectly extracted, with $0.99\omega_t$ matching $\omega_t$ and $-0.97\alpha^2 \sin(\theta_t)$ closely approximating the true $-\alpha^2 \sin(\theta_t)$. Given the complexity of the ME-Lotka-Volterra dataset, the extracted invariant functions $\hat{\mathbf{f}}_c$ are non-trivial and significantly outperform the full function $\hat{\mathbf{f}}$. On the ME-SIREpidemic dataset, the near-perfect NRMSE for the invariant state indicates that the invariant function must have been correctly extracted. However, although the expression for $\frac{dR_t}{dt}$ is correct, the extracted expressions for $\frac{dS_t}{dt}$ and $\frac{dI_t}{dt}$ do not precisely match the expected $\mathbf{f}_c$. Specifically, the coefficient from $\frac{dS_t}{dt}$ does not equal the inverse of the coefficient from $\frac{dI_t}{dt}$, though this discrepancy should be constant. These mismatches attribute to the limitations of PySR given the large number of variables and samples.[1] Future work could explore incorporating stronger inductive biases, similar to physics-informed machine learning (PIML) methods (Mouli et al., 2024; Yin et al., 2021b; Cranmer et al., 2020), to address these challenges. Please refer to Appx. E.3 for symbolic comparisons with all baseline.

## 6 Limitations

While this work provides a foundation for invariant function learning in dynamical systems, several limitations and opportunities for future exploration remain. These include extending the framework to more complex entanglements, exploring applications in PDE systems, strengthening theoretical guarantees with advanced error bounds, and developing comprehensive benchmarks. Additionally, broader applications such as generalizable physics learning and foundational model development represent exciting directions for further research. Please refer to Appx. I for more discussions.

## 7 Conclusion

In this work, we target addressing the challenge of the invariant mechanism discovery in ODE dynamical systems by extending invariant learning into function spaces. We introduce a new task, invariant function learning, which aims to extract the invariant dynamics across all environments with different environment-specific function forms. We design a causal analysis based disentanglement framework DIF to expose the underlying invariant functions. Additionally, we propose an invariant function learning principle with theoretical guarantees to optimize the framework and ensure effective invariant function discovery. Our experiments, including invariant trajectory validations, visualizations, ablation studies, and symbolic regression analyses, demonstrate the effectiveness of our method. Finally, as discussed in Sec. 6, the introduced invariant function learning task has wide application scenarios and many challenges remain to be addressed. We expect that our work will shed light on numerous future explorations in this field.

---

[1] Superior PySR explanations indicate great invariant function learning results, but effective invariant function learning results might not lead to good PySR explanations.

REPRODUCIBILITY STATEMENT

To make sure the reproducibility of this work, for our theoretical results, all assumptions and proofs are included in Appx. B. For the datasets, the system parameters, simulator expressions, and datasets visualizations are provided in Appx. C. For the model, we provide full details including all the training setup, architecture, objectives, and hyper-parameter searching spaces in Appx. D. The finalized code will be released upon acceptance.

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

# Appendix of Discovering Physics Laws of Dynamical Systems via Invariant Function Learning

## CONTENTS

# A  NOTATIONS

For the ease of reading, we providing a table including major notations below for reference.

Table 3: **Notation table**

| Notation | Explanation |
|---|---|
| $\boldsymbol{x}_t$ | Dynamical system states at time $t$ |
| $\mathcal{X}$ | Dynamical system state space |
| $T\mathcal{X}$ | Tangent state space |
| $\mathbb{R}$ | Real number space |
| $d$ | The number of state |
| $t$ | Time |
| $T_c$ | The first future time step |
| $X$ | A trajectory; A state matrix with T-step states |
| $X_p$ | Past states before time step $T_c$ |
| $X^c$ | An invariant trajectory |
| $\hat{X}, \hat{X}_p, \hat{X}^c$ | The predicted trajectory of $X$, $X_p$, and $X^c$ (by a model) |
| $\mathbf{X}, \mathbf{X}_p, \mathbf{X}^c$ | The matrix-valued random variable of $X$, $X_p$, and $X^c$ |
| $\hat{\mathbf{X}}, \hat{\mathbf{X}}_p, \hat{\mathbf{X}}^c$ | The predicted matrix-valued random variable of $X$, $X_p$, and $X^c$ |
| $f$ | A derivative function (of an underlying dynamical system) |
| $f_c$ | An invariant derivative function |
| $f_e$ | An environment derivative function |
| $\hat{f}, \hat{f}_c, \hat{f}_e$ | The predicted functions of $f$, $f_c$, and $f_e$ (by a model) |
| $\mathbf{f}, \mathbf{f}_c, \mathbf{f}_e$ | The derivative function random variable of $f$, $f_c$, and $f_e$ |
| $\hat{\mathbf{f}}, \hat{\mathbf{f}}_c, \hat{\mathbf{f}}_e$ | The predicted derivative function random variable of $f$, $f_c$, and $f_e$ |
| $\mathcal{F}$ | Function space/Functional vector space |
| $\hat{z}/\hat{z}_c/\hat{z}_e$ | A predicted full/invariant/environment hidden function representation |
| $\mathcal{Z}$ | Hidden function space/Hidden functional vector space |
| $h$ | A hypernetwork |
| $\mathcal{H}$ | Hypernetwork function space |
| $p(\cdot)$ | A probability distribution over a random variable |
| $I(\cdot;\cdot)$ | Mutual information between random variables |
| $H(\mathbf{X})$ | Shannon entropy of the matrix-valued random variable $\mathbf{X}$ |
| $\mathbb{E}_{\mathbf{X}\sim P}[f(X)]$ or $\mathbb{E}f(X)$ | Expectation of $f(X)$ with respect to $p(\mathbf{X})$ |
| $\mathbf{X} \sim P$ | Random variable $\mathbf{X}$ has distribution $P$ |
| $X \sim p(\mathbf{X})$ | $X$ is sampled from distribution $p(\mathbf{X})$ |
| $\mathbb{E}_{\mathbf{X}\sim P}[f(\mathbf{X})]$ or $\mathbb{E}f(\mathbf{X})$ | Expectation of $f(\mathbf{X})$ with respect to $p(\mathbf{X})$ |
| $\{\cdot\}$ | An assignment/A substitution rule |
| $\{\alpha \rightarrow\}$ | A symbol without an assigned value |
| $\{\alpha \rightarrow a\}$ | A symbol with an assigned value $a$ |

# B  INVARIANT FUNCTION LEARNING FOUNDATION

## B.1  STRUCTURAL CAUSAL MODEL

In this section, we discuss the trajectory generation process under the Structural Causal Model (SCM) assumption in Fig. 6. To begin with, this SCM is a directed acyclic graph (DAG) with the following components:

- Exogenous variables $U = \{c, e, \mathbf{X}_0, \epsilon_c, \epsilon_e, \epsilon_p, \epsilon\}$ are not caused by any variables within the model and are from their own independent distributions. Here $\epsilon_c, \epsilon_e, \epsilon_p, \epsilon$ are noise terms introduced during the function generation and trajectory integral.

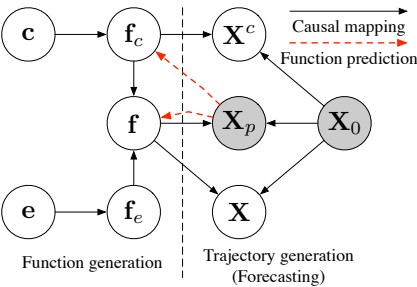

Figure 6: Structural causal model.

- Endogenous variables $V = \{f_c, f_e, f, \mathbf{X}^c, \mathbf{X}_p, \mathbf{X}\}$ are caused by the causal mappings within the model.
- Structural equations define the direct causation in the model.
  - $f_c := g_c(\mathbf{c}, \epsilon_c)$
  - $f_e := g_e(\mathbf{e}, \epsilon_e)$
  - $f := g_{comp}(f_c, f_e)$
  - $\mathbf{X}^c := g_{int}(f_c, \mathbf{X}_0) + \epsilon$
  - $\mathbf{X}_p := g_{int}^{T_c}(f, \mathbf{X}_0) + \epsilon_p$
  - $\mathbf{X} := g_{int}(f, \mathbf{X}_0) + \epsilon$

  where $g_{int}$ is a $T$-step integrator, while $g_{int}^{T_c}$ only applies integral for $T_c$ steps. While $g_c$, $g_e$, $g_{comp}$ are assumed to be unknown, $g_{int}$ is assumed to be an ideal integrator with an extra random noise $\epsilon$ to model the real world situations. Note that the cause of $\mathbf{X}$ can be written as $f_c$ and $f_e$ without side effects, so f is used for analytical purposes. Therefore, $g_{comp}$ is a conceptual function composition without introducing noises.

## B.2 PROOF OF INVARIANT FUNCTION LEARNING PRINCIPLE

**Theorem B.1** (Invariant Function Learning Principle 3.1). *Given the causal graph in Fig. 2 and the predicted function random variable $\hat{f}_c = h_{\theta_c}(\mathbf{X}_p)$, the true invariant function random variable is $f_c = h_{\theta_c^*}(\mathbf{X}_p)$, where $\theta_c^*$ is the solution to the following optimization problem:*

$$\theta_c^* = \arg\max_{\theta_c} I(h_{\theta_c}(\mathbf{X}_p); f|\mathbf{X}_0) \quad subject\ to \quad h_{\theta_c}(\mathbf{X}_p) \perp\!\!\!\perp \mathbf{e}, \tag{4}$$

*where $I(\cdot; \cdot)$ denotes mutual information, which quantifies the informational overlap between the predicted invariant function $h_{\theta_c}(\mathbf{X}_p)$ and the true full-dynamics function f.*

*Proof. Existence:*

We first prove the existence of a solution $\theta_c^*$ to the optimization problem, such that $f_c = h_{\theta_c^*}(\mathbf{X}_p)$. To establish this, we proceed by contradiction. Assume no such $\theta_c^*$ exists, implying that $I(f_c; f|\mathbf{X}_0)$ is not maximized. Then, there must exist some $f_c'$ such that $f_c' \perp\!\!\!\perp \mathbf{e}$ and $I(f_c'; f|\mathbf{X}_0) > I(f_c; f|\mathbf{X}_0)$. Given the mutual information expression $I(f_c; f|\mathbf{X}_0) = H(f|\mathbf{X}_0) - H(f|f_c, \mathbf{X}_0)$, this inequality implies:

$$H(f|f_c) > H(f|f_c'). \tag{5}$$

Since $\mathbf{X}_0$ is independent of f and $f_c$ (as per Fig. 2), and given that $f = g_f(f_c, f_e)$ implies $H(f|f_c, f_e) = 0$, we derive:

$$H(f|f_c) = H(f_e). \tag{6}$$

Similarly, for $f_c'$, we have:

$$H(f|f_c') \geq H(f_e|f_c'). \tag{7}$$

Combining these results with Eq. 5, we obtain:

$$H(f_e) > H(f_e|f_c'), \tag{8}$$

which contradicts the independence condition $f'_c \perp\!\!\!\perp e$, as this would require $H(f_e) = H(f_e|f'_c)$. Therefore, a solution $\theta^*_c$ exists, satisfying $f_c = h_{\theta^*_c}(\mathbf{X}_p)$.

*Uniqueness:* We now prove that for any solution $\theta^*_c$ of the optimization process, it holds that $f_c = h_{\theta^*_c}(\mathbf{X}_p)$.

We use a proof by contradiction. Assume that there exists another solution $f'_c \neq f_c$ that satisfies the independence constraint and achieves the maximum mutual information. By assumption, we have $H(e) = H(e|f'_c)$ and $I(f'_c; f|\mathbf{X}_0) = I(f_c; f|\mathbf{X}_0)$, which implies $H(f|f_c) = H(f|f'_c)$.

Since $f = g_f(f_c, f_e)$, we expand the entropy terms:

$$H(f|f_c) = H(f_c, f_e|f_c) = H(f_e), \tag{9}$$

and

$$H(f|f'_c) = H(f_c, f_e|f'_c) = H(f_c|f'_c) + H(f_e|f'_c) = H(f_c|f'_c) + H(f_e). \tag{10}$$

Substituting $H(f|f_c) = H(f|f'_c)$ into these equations, we find:

$$H(f_c|f'_c) = 0. \tag{11}$$

Based on this, we now aim to prove that $f_c = f'_c$. First, given $H(f'_c|f_c) \geq 0$, we need to show that $H(f'_c|f_c) = 0$. Assume that $H(f'_c|f_c) > 0$. In this case, $f'_c$ can determine $f_c$ while containing more information than $f_c$, all while remaining independent of $f_e$. This would imply that $H(f'|f) > 0$, where $f' = g_f(f', f_e)$, which would in turn affect the corresponding prediction, leading to $H(\mathbf{X}'|\mathbf{X}) > 0$. Such a situation would violate the MSE minimization condition.

Therefore, we must have both $H(f'_c|f_c) = 0$ and $H(f_c|f'_c) = 0$. This implies that $f_c$ and $f'_c$ are isomorphic functions, i.e., there exists a bijective function $g_b$ such that $f_c = g_b(f'_c)$.

Furthermore, since minimizing the MSE of $X$ is equivalent to minimizing the MSE of $\frac{dX}{dt}$ given $X_0$, this minimization ensures that, for the same $f_e$, $f_c(X) = f'_c(X)$ for all $X$. As a result, the bijective function $g_b$ must be the identity mapping, and we conclude that $f'_c = f_c$ in the support of $p(\mathbf{X})$.

Thus, for any solution $\theta^*_c$ of the optimization process, it follows that $f_c = h_{\theta^*_c}(\mathbf{X}_p)$.

$\square$

### B.3 PROOF OF ODE CROSS-ENTROPY MINIMIZATION

**Lemma B.2** (ODE cross-entropy minimization 3.2). *Given our forecasting model $p(\mathbf{X}|h_\theta(\mathbf{X}_p), \mathbf{X}_0)$, it follows that the cross-entropy minimization between the data distribution $p(\mathbf{X})$ and $p(\mathbf{X}|h_\theta(\mathbf{X}_p), \mathbf{X}_0)$ is equivalent to minimizing mean square error $\min_\theta \mathbb{E}_{X \sim p} \|X - \hat{X}\|_2^2$, where $\hat{X}$ is sampled from $p(\mathbf{X}|h_\theta(X_p), X_0)$.*

*Proof.* The forecasting optimization goal is to use our framework to approximate the data distribution $p(\mathbf{X})$, parameterized as $p(\mathbf{X}|h_\theta(\mathbf{X}_p), \mathbf{X}_0)$, where we apply the cross-entropy minimization, *i.e.*, $H(p(\mathbf{X}), p(\mathbf{X}|h_\theta(\mathbf{X}_p), \mathbf{X}_0)) = -\mathbb{E}_{\mathbf{X} \sim p}[\log p(X|h_\theta(X_p), X_0)]$. Furthermore, this negative log-likelihood optimization can be further reduced to the common mean squared error (MSE).

$$\min_\theta \mathbb{E}_{\mathbf{X} \sim p} \|X - \hat{X}\|_2^2, \tag{12}$$

where $\hat{X} \sim p(\mathbf{X}|h_\theta(X_p), X_0)$. Since the distribution $p(\mathbf{X}|h_\theta(X_p), X_0)$ is modeled as a Gaussian $\mathcal{N}\left(\mathbf{X}; g_{int}(h_\theta(X_p), X_0), \sigma^2 I\right)$ (Sec. 3.1), we have $\hat{X} = \mu + \epsilon$, where $\epsilon \sim \mathcal{N}\left(0, \sigma^2 I\right)$.

$$\min_\theta -\mathbb{E}_{\mathbf{X} \sim p}[\log p(X|h_\theta(X_p), X_0)]$$

$$= \min_\theta \mathbb{E}_{\mathbf{X} \sim p} \left[ \frac{n}{2} \log(2\pi\sigma^2) + \frac{1}{2\sigma^2} \|X - \mu\|_2^2 \right] \tag{13}$$

$$= \min_\theta \mathbb{E}_{\mathbf{X} \sim p} \left[ \frac{1}{2\sigma^2} \|X - \mu\|_2^2 \right] + \frac{n}{2} \log(2\pi\sigma^2)$$

Since $\frac{n}{2}\log(2\pi\sigma^2)$ is a constant, we ignore it in the minimization process.

$$
\min_\theta \mathbb{E}_{\mathbf{X}\sim p}\left[\frac{1}{2\sigma^2}\|X-\mu\|_2^2\right] + \frac{n}{2}\log(2\pi\sigma^2)
$$

$$
=\min_\theta \mathbb{E}_{\mathbf{X}\sim p}\left[\frac{1}{2\sigma^2}\|X-(\hat{X}-\epsilon)\|_2^2\right]
$$

$$
=\min_\theta \mathbb{E}_{\mathbf{X}\sim p}\left[\frac{1}{2\sigma^2}\|X-\hat{X}+\epsilon\|_2^2\right] \tag{14}
$$

$$
=\min_\theta \mathbb{E}_{\mathbf{X}\sim p}\left[\frac{1}{2\sigma^2}(\|X-\hat{X}\|_2^2+\|\epsilon\|_2^2+2(X-\hat{X})^T\epsilon)\right]
$$

$$
=\min_\theta \mathbb{E}_{\mathbf{X}\sim p}\left[\frac{1}{2\sigma^2}\|X-\hat{X}\|_2^2\right] + \frac{1}{2\sigma^2}\mathbb{E}_\epsilon[\|\epsilon\|_2^2] + \frac{1}{\sigma^2}\mathbb{E}_{\mathbf{X}\sim p}(X-\hat{X})^T\mathbb{E}_\epsilon[\epsilon]
$$

Here, $\frac{1}{2\sigma^2}\mathbb{E}_{\mathbf{X}\sim p}[\|\epsilon\|_2^2]$ is a constant; $\mathbb{E}_{\mathbf{X}\sim p}\left[\frac{1}{\sigma^2}(X-\hat{X})^T\epsilon\right]=\frac{1}{\sigma^2}\mathbb{E}_{\mathbf{X}\sim p}(X-\hat{X})^T\mathbb{E}_\epsilon[\epsilon]=0$ since $\epsilon$ is independently sampled with a zero mean. Therefore,

$$
\min_\theta \mathbb{E}_{\mathbf{X}\sim p}\left[\frac{1}{2\sigma^2}\|X-\hat{X}\|_2^2\right] + \frac{1}{2\sigma^2}\mathbb{E}_\epsilon[\|\epsilon\|_2^2] + \frac{1}{\sigma^2}\mathbb{E}_{\mathbf{X}\sim p}(X-\hat{X})^T\mathbb{E}_\epsilon[\epsilon]
$$

$$
=\min_\theta \mathbb{E}_{\mathbf{X}\sim p}\left[\frac{1}{2\sigma^2}\|X-\hat{X}\|_2^2\right] \tag{15}
$$

$$
=\min_\theta \mathbb{E}_{\mathbf{X}\sim p}\left[\|X-\hat{X}\|_2^2\right]
$$

This reduction connects the information theory and the practical MSE optimization, which further helps us to transform the mutual information maximization into a similar MSE optimization below. $\square$

### B.4 Proof of ODE conditional mutual information maximization

**Proposition B.3** (Proof of ODE conditional mutual information maximization 3.3). *Given forecasting model $p(\mathbf{X}|h_{\theta_c}(X_p),X_0)$, it follows that the conditional mutual information maximization $\max_{\theta_c} I(h_{\theta_c}(\mathbf{X}_p);\mathrm{f}|\mathbf{X}_0)$ is equivalent to minimizing mean square error $\min_{\theta_c} \mathbb{E}_{\mathbf{X}\sim P}\|X-\hat{X}^c\|_2^2$, where $\hat{X}^c$ is the predicted trajectory sampled from $p(\mathbf{X}|h_{\theta_c}(X_p),X_0)$.*

*Proof.* According to Lemma 3.2, we can reduce a negative log-likelihood minimization $\min_{\theta_c} -\mathbb{E}_{\mathbf{X}\sim p}\log p(X|h_{\theta_c}(X_p),X_0)$ to $\min_{\theta_c} \mathbb{E}_{\mathbf{X}\sim P}\|X-\hat{X}^c\|_2^2$.

Therefore, we only need to prove the equivalence between the $-\mathbb{E}_{\mathbf{X}\sim p}\log p(X|h_{\theta_c}(X_p),X_0)$ minimization and the $I(h_{\theta_c}(\mathbf{X}_p);\mathrm{f}|\mathbf{X}_0)$ maximization. It follows that

$$
\max_{\theta_c} I(h_{\theta_c}(\mathbf{X}_p);\mathrm{f}|\mathbf{X}_0)
$$

$$
=\max_{\theta_c} H(\mathrm{f}|\mathbf{X}_0) - H(\mathrm{f}|h_{\theta_c}(\mathbf{X}_p),\mathbf{X}_0). \tag{16}
$$

Since $H(\mathrm{f}|\mathbf{X}_0)$ is a constant, we ignore it in the maximization process and obtain

$$
\max_{\theta_c} H(\mathrm{f}|\mathbf{X}_0) - H(\mathrm{f}|h_{\theta_c}(\mathbf{X}_p),\mathbf{X}_0)
$$

$$
=\max_{\theta_c} -H(\mathrm{f}|h_{\theta_c}(\mathbf{X}_p),\mathbf{X}_0) \tag{17}
$$

$$
=\min_{\theta_c} H(\mathrm{f}|h_{\theta_c}(\mathbf{X}_p),\mathbf{X}_0).
$$

Since $\mathbf{X} = g_{int}(\mathrm{f},\mathbf{X}_0) + \epsilon$ where $\epsilon$ is an indepedent random noise, $H(\mathbf{X}|h_{\theta_c}(\mathbf{X}_p),\mathbf{X}_0,\mathrm{f}) = H(\epsilon|h_{\theta_c}(\mathbf{X}_p)) = H(\epsilon)$. Since given specific $\theta_c$, $H(h_{\theta_c}(\mathbf{X}_p)|\mathbf{X}) = 0$, with the conditional entropy

chain rule, it follows that

$$\min_{\theta_c} H(f|h_{\theta_c}(\mathbf{X}_p), \mathbf{X}_0)$$

$$= \min_{\theta_c} H(f, \mathbf{X}|h_{\theta_c}(\mathbf{X}_p), \mathbf{X}_0) - H(\mathbf{X}|h_{\theta_c}(\mathbf{X}_p), \mathbf{X}_0, f)$$

$$= \min_{\theta_c} H(f, \mathbf{X}|h_{\theta_c}(\mathbf{X}_p), \mathbf{X}_0) - H(\epsilon)$$

$$= \min_{\theta_c} H(f, \mathbf{X}|h_{\theta_c}(\mathbf{X}_p), \mathbf{X}_0)$$

$$= \min_{\theta_c} H(f|h_{\theta_c}(\mathbf{X}_p), \mathbf{X}_0, \mathbf{X}) + H(\mathbf{X}|h_{\theta_c}(\mathbf{X}_p), \mathbf{X}_0)$$  (18)

$$= \min_{\theta_c} H(f|\mathbf{X}_0, \mathbf{X}) + H(\mathbf{X}|h_{\theta_c}(\mathbf{X}_p), \mathbf{X}_0)$$

$$= \min_{\theta_c} H(\mathbf{X}|h_{\theta_c}(\mathbf{X}_p), \mathbf{X}_0)$$

$$= \min_{\theta_c} -\mathbb{E}_{\mathbf{X}, \mathbf{x}_p, \mathbf{x}_0 \sim p} \log p(X|h_{\theta_c}(X_p), X_0)$$

$$= \min_{\theta_c} -\mathbb{E}_{\mathbf{X} \sim p} \log p(X|h_{\theta_c}(X_p), X_0)$$

where $H(f|\mathbf{X}_0, \mathbf{X})$ is a constant. Here, we finish building the equivalence between the function conditional mutual information maximization and the trajectory negative log-likelihood minimization. Using the proof in Lemma 3.2, our final optimization goal can be reduced to

$$\min_{\theta_c} \mathbb{E}_{\mathbf{X} \sim P} \|X - \hat{X}^c\|_2^2.$$  (19)

$\square$

## B.5 THEORETICAL JUSTIFICATION FOR ADVERSARIAL TRAINING

To incorporate the independence constraint, we enforce the condition $\hat{f}_c \perp\!\!\!\perp e$, where $\hat{f}_c = h_{\theta_c}(\mathbf{X}_p)$ is the predicted function random variable, not a realization. Since $\hat{f}_c \perp\!\!\!\perp e$ is equivalent to $I(e; \hat{f}_c) = 0$, and $I(e; \hat{f}_c) \geq 0$, the objective $I(e; \hat{f}_c)$ reaches its minimum when and only when $\hat{f}_c \perp\!\!\!\perp e$. This leads us to define minimizing $I(e; \hat{f}_c)$ as our training criterion.

**Definition 1.** The environment independence training criterion is

$$\theta_c^* = \arg\min_{\theta_c} I(e; \hat{f}_c).$$  (20)

This training criterion can be used directly, since the mutual information $I(e; \hat{f}_c) = \mathbb{E}\left[\log \frac{P(e|\hat{f}_c)}{P(e)}\right]$ while $P(e|\hat{f}_c)$ is unknown. Following Proposition 1 in GAN Goodfellow et al. (2020), we introduce an optimal discriminator $g_\phi : \mathcal{F} \mapsto \mathcal{E}$ with parameters $\phi$ to approximate the unknown $P(e|\hat{f}_c)$ as $P_\phi(e|\hat{f}_c)$, minimizing the negative log-likelihood $-\mathbb{E}\left[\log P_\phi(e|\hat{f}_c)\right]$. We then have the following two propositions:

**Proposition B.4.** *For $\theta_c$ fixed, the optimal discriminator $\phi$ is*

$$\phi^* = \arg\min_{\phi} -\mathbb{E}\left[\log P_\phi(e|\hat{f}_c)\right].$$  (21)

This proposition can be proved straightforwardly by applying the cross-entropy training criterion.

**Proposition B.5.** *Denoting KL-divergence as* $\mathrm{KL}[\cdot\|\cdot]$, *for $\theta_c$ fixed, the optimal discriminator $\phi$ is $\phi^*$, such that*

$$\mathrm{KL}\left[P(e|\hat{f}_c)\|P_{\phi^*}(e|\hat{f}_c)\right] = 0.$$  (22)

*Proof.* Given a fixed $\theta_c$, both $I(e; \hat{f}_c)$ and $H(e)$ are constants. Therefore, we have:

$$\phi^* = \arg\min_{\phi} -\mathbb{E}\left[\log P_\phi(e|\hat{f}_c)\right]$$

$$= \arg\min_{\phi} I(e; \hat{f}_c) - \mathbb{E}\left[\log P_\phi(e|\hat{f}_c)\right] - H(e)$$  (23)

$$= \arg\min_{\phi} \mathrm{KL}\left[P(e|\hat{f}_c)\|P_\phi(e|\hat{f}_c)\right].$$

Thus, minimizing the negative log-likelihood of $P_\phi(\mathbf{e}|\hat{\mathbf{f}}_c)$ is equivalent to minimizing the KL divergence between $P(\mathbf{e}|\hat{\mathbf{f}}_c)$ and its approximation $P_\phi(\mathbf{e}|\hat{\mathbf{f}}_c)$. Since KL divergence is bounded by 0, we have $\mathrm{KL}\left[P(\mathbf{e}|\hat{\mathbf{f}}_c)\|P_{\phi^*}(\mathbf{e}|\hat{\mathbf{f}}_c)\right] = 0$. This concludes the proof. $\qquad\square$

With these propositions, the mutual information can be computed with the help of the optimal discriminator $\phi^*$. According to Proposition B.5, we have:

$$
\begin{aligned}
I(\mathbf{e};\hat{\mathbf{f}}_c) &= \mathbb{E}\left[\log P_{\phi^*}(\mathbf{e}|\hat{\mathbf{f}}_c)\right] + H(\mathbf{e}) + \mathrm{KL}\left[P(\mathbf{e}|\hat{\mathbf{f}}_c)\|P_{\phi^*}(\mathbf{e}|\hat{\mathbf{f}}_c)\right] \\
&= \mathbb{E}\left[\log P_{\phi^*}(\mathbf{e}|\hat{\mathbf{f}}_c)\right] + H(\mathbf{e}) + 0.
\end{aligned}
\tag{24}
$$

Thus, by disregarding the constant $H(\mathbf{e})$, the training criterion becomes:

$$
\begin{aligned}
\theta_c^* &= \arg\min_{\theta_c} I(\mathbf{e};\hat{\mathbf{f}}_c) \\
&= \arg\min_{\theta_c} \mathbb{E}\left[\log P_{\phi^*}(\mathbf{e}|\hat{\mathbf{f}}_c)\right] \\
&= \arg\min_{\theta_c}\left\{\max_\phi \mathbb{E}\left[\log P_\phi(\mathbf{e}|\hat{\mathbf{f}}_c)\right]\right\},
\end{aligned}
\tag{25}
$$

where $P_\phi(\mathbf{e}|\hat{\mathbf{f}}_c)$ is the probability modeling of $g_\phi$. Therefore, the log-likelihood adversarial training can enforce the independence $h_{\theta_c}(\mathbf{X}_p) \perp\!\!\!\perp \mathbf{e}$.

## C  DATASETS

### C.1  BASIC SETUP

We conduct experiments on the proposed three multi-environment datasets ME-Pendulum, ME-Lotka-Volterra, and ME-SIREpidemic. Each of these datasets includes 1000 samples, where 800 and 200 samples are assigned to training set and test set, respectively. Each training set has 4 environments where 200 samples are generated in each environment. Each sample is observed over 10 units of time, and each time is discretized by regularly-spaced discrete time steps from $t_0$ to $t_T$, where $T = 99$, *i.e.*, there are 100 time intervals of 0.1 unit of time each. A sample generation process is controlled by a set of ODEs with common parameters $\mathbf{W}^c \sim \mathcal{U}(W_{low}^c, W_{high}^c)$ and environment-specific parameters $\mathbf{W}^e \sim \mathcal{U}(W_{low}^e, W_{high}^e)$ sampled from their uniform distributions, *e.g.*, in the pendulum system, $\mathbf{W}^c = \{\boldsymbol{\alpha}\}$ and $\mathbf{W}^e = \{\boldsymbol{\rho}\}$, where $\boldsymbol{\alpha} \sim \mathcal{U}(\alpha_{low}, \alpha_{high})$ and $\boldsymbol{\rho} \sim \mathcal{U}(\rho_{low}, \rho_{high})$. Note that each environment has its specific function form with its environment-specific parameters $\mathbf{W}^e$.

The environment split is the same in the test set. The only difference is that in the test set each sample $X$ has one additional prediction target $X^c$ with only the invariant dynamics. For example, the invariant trajectory $X^c$ of the one generated by $-\alpha^2 \sin\theta_t - \rho\omega_t$ will be created by $-\alpha^2 \sin\theta_t$.

### C.2  ME-PENDULUM

ME-Pendulum is motivated by the DampedPendulum system (Yin et al., 2021b). The state $X_t = [\theta_t, \omega_t] \in \mathbb{R}^2$ are the angle and angular velocity of the pendulum at time $t$, where we have $\mathbf{W}^c = \{\boldsymbol{\alpha}\}$, $\mathbf{W}^e = \{\boldsymbol{\rho}\}$, and $T_C = \frac{T}{3}$. The underlying invariant ODE is $\frac{d\theta_t}{dt} = \omega_t, \frac{d\omega_t}{dt} = -\boldsymbol{\alpha}^2 \sin(\theta_t)$. As shown in Tab. 4, this invariant ODE is entangled with different environmental factors, forming four environments, namely, *damped*, *powered*, *spring*, *air*.

### C.3  ME-LOTKA-VOLTERRA

Motivated by the Lotka-Volterra system (Ahmad, 1993), the state $X_t = [p_t, q_t] \in \mathbb{R}^2$ are the population of preys and predators at time $t$, where we have $\mathbf{W}^c = \{\boldsymbol{\alpha}, \boldsymbol{\beta}, \boldsymbol{\gamma}, \boldsymbol{\delta}\}$, $\mathbf{W}^e = \{\boldsymbol{\alpha}', \boldsymbol{\beta}', \boldsymbol{\gamma}', \boldsymbol{\delta}'\}$, and $T_C = \frac{T}{2}$. The underlying invariant ODEs are $\frac{dp}{dt} = \boldsymbol{\alpha}p - \boldsymbol{\beta}pq, \frac{dq}{dt} = \boldsymbol{\delta}pq - \boldsymbol{\gamma}q$. As shown in Tab. 5, these invariant ODEs are entangled in 4 environments, *i.e.*, *save*, *fight*, *resource*, *omnivore*. The save environment ODEs simulate the decrease of food wastage along with the increase of predators. The fight environment ODEs simulate the decrease of hunting efficiency along with the increase of

Table 4: **ME-Pendulum ODEs** with $\theta_0 \sim \mathcal{U}(0, \frac{\pi}{2})$ and $\omega_0 \sim \mathcal{U}(-1, 0)$.

| Environment | ODE for $\theta_t$ | ODE for $\omega_t$ | Distribution of Parameters |
|---|---|---|---|
| Damped | $\frac{d\theta_t}{dt} = \omega_t$ | $\frac{d\omega_t}{dt} = -\boldsymbol{\alpha}^2 \sin(\theta_t) - \boldsymbol{\rho}\omega_t$ | |
| Powered | $\frac{d\theta_t}{dt} = \omega_t$ | $\frac{d\omega_t}{dt} = -\boldsymbol{\alpha}^2 \sin(\theta_t) + \boldsymbol{\rho}\frac{\omega_t}{|\omega_t|}$ | $\boldsymbol{\alpha} \sim \mathcal{U}(1.0, 2.0)$ |
| Spring | $\frac{d\theta_t}{dt} = \omega_t$ | $\frac{d\omega_t}{dt} = -\boldsymbol{\alpha}^2 \sin(\theta_t) - \boldsymbol{\rho}\theta_t$ | $\boldsymbol{\rho} \sim \mathcal{U}(0.2, 0.4)$ |
| Air | $\frac{d\theta_t}{dt} = \omega_t$ | $\frac{d\omega_t}{dt} = -\boldsymbol{\alpha}^2 \sin(\theta_t) - \boldsymbol{\rho}|\omega_t|\omega_t$ | |
| Invariant | $\frac{d\theta_t}{dt} = \omega_t$ | $\frac{d\omega_t}{dt} = -\boldsymbol{\alpha}^2 \sin(\theta_t)$ | |

predators. The resource environment ODEs limit the increase rate of the prey population. In the omnivore environment ODEs, the predators are omnivores that can build the population without preys under certain resource limits.

We plot the trajectories $X$ in the training set, and the invariant trajectories $X^c$ in the test set in Fig. 7.

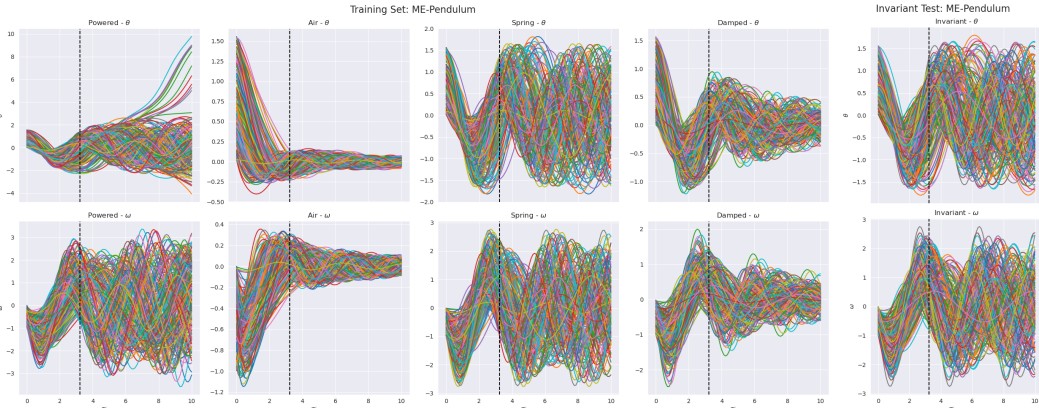

Figure 7: **ME-Pendulum trajectories.**

Table 5: **ME-Lotka-Volterra ODEs** with $p_0 \sim \mathcal{U}(1000, 2000)$ and $q_0 \sim \mathcal{U}(10, 20)$.

| Environment | ODE for $p_t$ | ODE for $q_t$ | Distribution of Parameters |
|---|---|---|---|
| Save | $\frac{dp}{dt} = \boldsymbol{\alpha}p - \boldsymbol{\beta}pq - \boldsymbol{\beta}'pq \cdot 10\exp\left(-\frac{q}{10}\right)$ | $\frac{dq}{dt} = \boldsymbol{\delta}pq - \boldsymbol{\gamma}q$ | $\boldsymbol{\alpha}, \boldsymbol{\alpha}' \sim \mathcal{U}(1.2, 2.4)$ |
| Fight | $\frac{dp}{dt} = \boldsymbol{\alpha}p - \boldsymbol{\beta}pq$ | $\frac{dq}{dt} = \boldsymbol{\delta}pq + \boldsymbol{\delta}'pq \cdot 10\exp\left(-\frac{q}{10}\right) - \boldsymbol{\gamma}q$ | $\boldsymbol{\beta}, \boldsymbol{\beta}' \sim \mathcal{U}(6e-2, 1.2e-1)$ |
| Resource | $\frac{dp}{dt} = \boldsymbol{\alpha}p - \boldsymbol{\alpha}'\frac{p^2}{2000} - \boldsymbol{\beta}pq$ | $\frac{dq}{dt} = \boldsymbol{\delta}pq - \boldsymbol{\gamma}q$ | $\boldsymbol{\gamma}, \boldsymbol{\gamma}' \sim \mathcal{U}(0.48, 0.96)$ |
| Omnivore | $\frac{dp}{dt} = \boldsymbol{\alpha}p - \boldsymbol{\beta}pq$ | $\frac{dq}{dt} = \boldsymbol{\delta}pq + 20\boldsymbol{\gamma}'\left(1 - \frac{q}{100}\right) - \boldsymbol{\gamma}q$ | $\boldsymbol{\delta}, \boldsymbol{\delta}' \sim \mathcal{U}(4.8e-4, 9.6e-4)$ |
| Invariant | $\frac{dp}{dt} = \boldsymbol{\alpha}p - \boldsymbol{\beta}pq$ | $\frac{dq}{dt} = \boldsymbol{\delta}pq - \boldsymbol{\gamma}q$ | |

We plot the trajectories $X$ in the training set, and the invariant trajectories $X^c$ in the test set in Fig. 8.

### C.4 ME-SIREPIDEMIC

In the SIREpidemic (Wang et al., 2021) system, the states $X_t = [S_t, I_t, R_t] \in \mathbb{R}^3$ are the susceptible, infected, and recovered individuals at time $t$, respectively. In this adapted ME-SIREpidemic system, we have $\mathbf{W}^c = \{\boldsymbol{\beta}\}$, $\mathbf{W}^e = \{\boldsymbol{\gamma}\}$, and $T_c = \frac{T}{2}$. The underlying invariant ODEs are $\frac{dS}{dt} = -\beta\frac{SI}{S+I+R}, \frac{dI}{dt} = \beta\frac{SI}{S+I+R}, \frac{dR}{dt} = 0$, where we only care about the $S$ to $I$ transformation relationship. As shown in Tab. 6, we introduce 4 environments, *origin*, *enlarge*, *loop*, *negative*. These four environments describe four different models, where some of them are only for math modeling. The origin environment is the same as the original SIREpidemic model. The enlarge environment ODEs expand the epidemic range. The loop environment ODEs include deaths and second-time infections. The negative environment ODEs is a pure math model allowing negative numbers.

We plot the trajectories $X$ in the training set, and the invariant trajectories $X^c$ in the test set in Fig. 9.

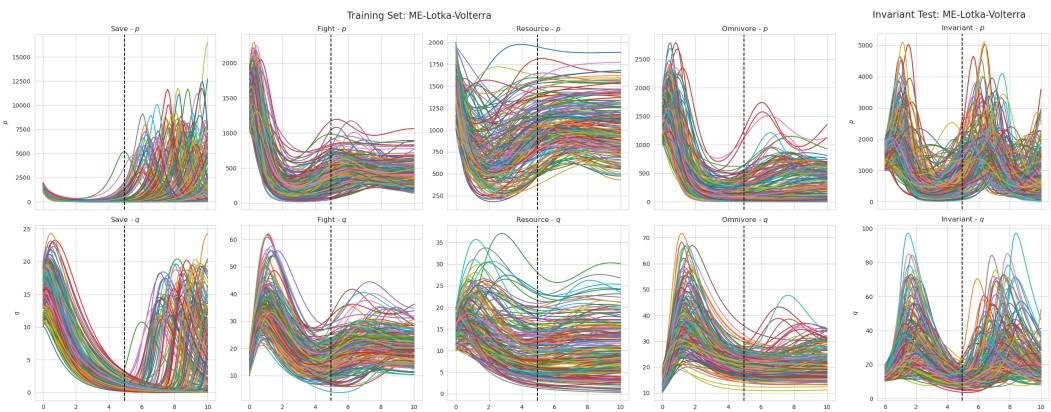

Figure 8: **ME-Lotka-Volterra trajectories.**

Table 6: **ME-SIREpidemic ODEs** with $S_0 \sim \mathcal{U}(9, 10)$, $I_0 \sim \mathcal{U}(1, 5)$, and $R_0 = 0$.

| Environment | ODE for $S_t$ | ODE for $I_t$ | ODE for $R_t$ | Distribution of Parameters |
|---|---|---|---|---|
| Origin | $\frac{dS}{dt} = -\beta \frac{SI}{S+I+R}$ | $\frac{dI}{dt} = \beta \frac{SI}{S+I+R} - \gamma I$ | $\frac{dR}{dt} = \gamma I$ | |
| Enlarge | $\frac{dS}{dt} = -\beta \frac{SI}{S+I+R} + \gamma I$ | $\frac{dI}{dt} = \beta \frac{SI}{S+I+R} - \gamma I$ | $\frac{dR}{dt} = \gamma I$ | $\boldsymbol{\beta} \sim \mathcal{U}(4, 8)$ |
| Loop | $\frac{dS}{dt} = -\beta \frac{SI}{S+I+R} + \gamma I + \gamma R$ | $\frac{dI}{dt} = \beta \frac{SI}{S+I+R} - 2\gamma I$ | $\frac{dR}{dt} = \gamma I - \gamma R$ | $\boldsymbol{\gamma} \sim \mathcal{U}(0.4, 0.8)$ |
| Negative | $\frac{dS}{dt} = -\beta \frac{SI}{S+I+R}$ | $\frac{dI}{dt} = \beta \frac{SI}{S+I+R} + \gamma \log I$ | $\frac{dR}{dt} = -\gamma \log I$ | |
| Invariant | $\frac{dS}{dt} = -\beta \frac{SI}{S+I+R}$ | $\frac{dI}{dt} = \beta \frac{SI}{S+I+R}$ | $\frac{dR}{dt} = 0$ | |

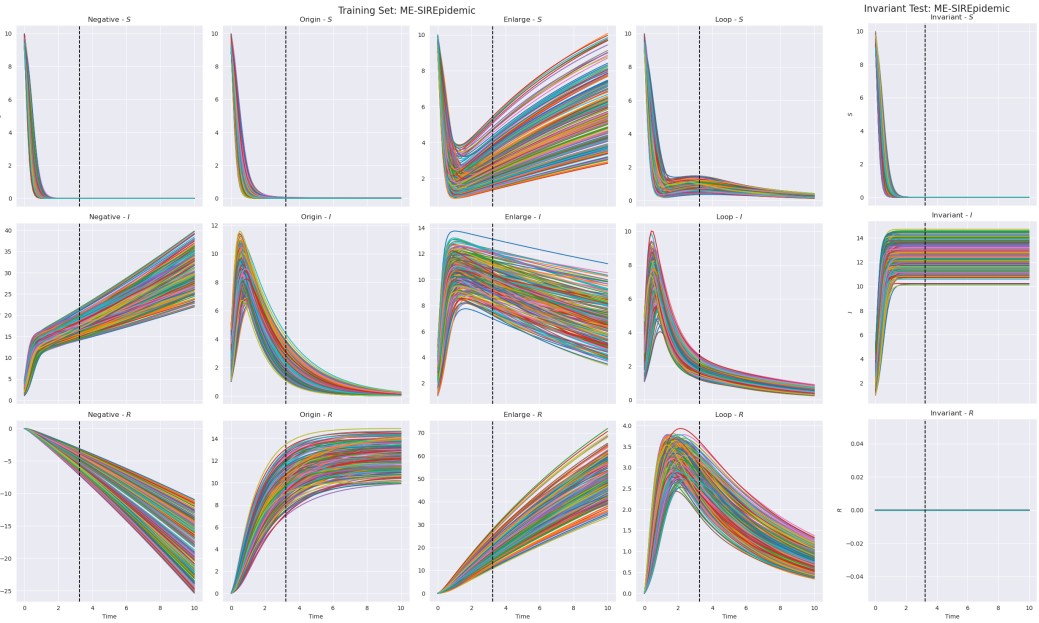

Figure 9: **ME-SIREpidemic trajectories.**

## D  EXPERIMENTAL DETAILS

We conduct experiments on 800-sample training sets with a training batch size of 32, which leads to 25 iterations per epoch. For each run, we optimize the neural network with 2,000 epochs, which is equivalent to 50,000 iterations. Given fixed learning iterations, the learning rate is selected from $\mathcal{U}(1e - 4, 1e - 3)$.

## D.1 BASELINES

In our experiments, we design four adapted baselines, since this new task has never been explored before. The selection of baselines is based on the following two questions.

- As invariant learning has been successfully applied in many representation learning tasks, can general invariant learning principle still work for invariant function learning?

- Meta-learning techniques has been well designed to solved problems in dynamical systems due to their quick adaptation characteristics. However, there is no evidence that the quick adaptable functions are the invariant function that shares across environments. Are they?

For the first question, since our method is based on enforcing independence which shares a similar philosophy as domain adversarial neural network Ganin et al. (2016), we consider other invariant methods going different ways. There are many methods well-known in the field of invariant learning, but considering our invariant function learning formulation, we found the discovery of invariant function requires independence that is different from the most typical "well performed across all environments" invariant learning requirements. Therefore, as a validation of our guess, we adapt two widely used and known invariant learning methods, IRM Arjovsky et al. (2019) and VREx Krueger et al. (2021). These two techniques are directly applied to our framework with the same architecture and their typical hyper-parameters searching spaces:

- $\lambda_{irm} \sim \mathcal{U}(1e-2, 1e2)$
- $\lambda_{vrex} \sim \mathcal{U}(1e-1, 1e3)$

As shown in our result plot 4, the results are not surprising. That means that the function can perform well across multiple environments is not the invariant function.

For the second question, our initial guess is that meta-learning is very sensitive to the distribution of the training set. If certain pattern exists in multiple environments (not all), the meta-learning methods are prone to capture it as a part of the meta-function, which satisfies their quick adaptation goals. In our experiments, we choose MAML Finn et al. (2017) and CoDA Kirchmeyer et al. (2022) as our adapted baselines. CoDA is adapted since it uses hypernetwork as a full network adaptor which is similar to our framework. However, CoDA is applied on coefficient-environments and requires test-time adaptation without a trajectory encoder, leading to significant architecture differences. Therefore, we apply CoDA as meta-learning techniques focusing on its low-dimension (2-dimension) environment representation and regularization. MAML is selected as the most typical meta-learning baseline, which does not require the use of hypernetwork, and only learns a meta function used to predict invariant trajectories. Their typical hyper-parameters searching spaces are shown as follows.

- $\lambda_{coda} \sim \mathcal{U}(1e-5, 1e-3)$
- $\lambda_{maml} \sim \mathcal{U}(1e-3, 1)$      (Meta learning rate)

## D.2 DISENTANGLEMENT OF INVARIANT FUNCTION SETUP

### D.2.1 ARCHITECTURE

**Transformer.** For the trajectory encoder in our hypernetwork, we apply a 6-layer 8-head 256-dimension FFN transformer Vaswani (2017) with frequency positional encoding Gehring et al. (2017). We tried different architectures like GRUs Cho (2014), but the transformer encoder can provide the best in-domain test performance easily. We also sweeped to the depth, width, and number of heads, and found that 6-layer 8-head 256-dimension FFN transformer is strong enough for our ODE systems without making training difficult.

**Function embedding.** In the hidden function embedding space, we select the function embedding dimension to be 32 or 64, while these two selections perform quite similar. For the MLPs used to disentangle and decode hidden function embedding, we use 3-layer MLPs with ReLU as the activations. While for the decoder, the last layer projects the hidden function embedding to parameterized function space $\mathbb{R}^m$ where $m$ is the number of parameters in the derivative neural network.

**Derivative function network.** The derivative neural network is a 4-layer or 5-layer MLP with width 16 or 32, which takes $X_t \in \mathbb{R}^d$ as input and output $\frac{dX_t}{dt} \in \mathbb{R}^d$. This neural network is transformed to be a functional in our implementation.

**Discriminator.** Our discriminator is a 3 to 6 layer MLP with width 64 or 128. The size of discriminator can be easily chosen since the main goal of it is to discriminate the environment information in the hidden function space. Therefore, the simplest way to filter non-qualified discriminators is using the prediction entropy of $P_\phi(\mathrm{e}|\hat{\mathrm{f}}_e)$, since $\hat{\mathrm{f}}_e$ should contain rich environment information, different to the prediction from $\hat{\mathrm{f}}_c$. Our experiments also validate that the failure to distinguish $\hat{\mathrm{f}}_e$ will always cause the failure of invariant function discovery, which is natural, since the adversarial training is based on the optimal discriminators Goodfellow et al. (2020).

Note that for all our MLPs, we apply one LayerNorm before each activation.[2]

### D.2.2    TRAINING OBJECTIVES

We restate our three additional strategies here. Firstly, the adversarial training of $\hat{\mathrm{f}}_c$ will cause the loss of environment information in $\hat{\mathrm{f}}_c$, leading to the training difficulty of the discriminator; therefore, this discriminator is not only trained on $\hat{\mathrm{f}}_c$ but also on $\hat{\mathrm{f}}_e$ with the same hyper-parameter $\lambda_{dis}$, *i.e.*, $\min_{\phi,\theta} -\mathbb{E}_{\mathbf{X},\mathrm{e}\sim P}\left[\log P_\phi(e|\hat{f}_e)\right]$. Secondly, instead of using the large $\hat{f}_c$ and $\hat{f}_e$ as the input of the discriminator, we input the the corresponding embeddings $\boldsymbol{z}_c$ and $\boldsymbol{z}_e$. Thirdly, to avoid the use of a numerical or neural integrator which causes long training time, we follow Mouli et al. (2024) to fit derivatives only. That is, instead of using the inference forecaster $p(X|h_{\theta_c}(X_p), X_0)$, we calculate the derivatives of $X$ using $\hat{f}$ and $\hat{f}_c$, and replace the MSE over trajectory matrices with the MSE over derivative matrices. Note that this modification only eliminates the use of integrator for stability during training and thus does not affect our analysis and optimization goal.

We introduce our hyper-parameter searching space as follows.

- $\lambda_c \sim \mathcal{U}(1e-7, 1e-4)$

- $\lambda_{dis} \sim \mathcal{U}(1e-1, 1)$

- $\lambda'_{adv} \sim \mathcal{U}(1e2, 1e6)$

- $\lambda_{adv} = \lambda_c \cdot \lambda'_{adv}$

The most critical hyper-parameters are $\lambda_c$ and $\lambda_{adv}$ which control the information overlap between $\mathrm{f}_c$ and f. Conceptually, $\lambda_c$ controls the conditional mutual information (MI) maximization in our invariant function learning principle, while $\lambda_{adv}$ enforces the independence constraint. The intensity of the independence enforcing $\lambda_{adv}$ is dependent on the intensity of MI maximization $\lambda_c$; thus, we set $\lambda_{adv}$ according to $\lambda_c$, leading to $\lambda_{adv} = \lambda_c \cdot \lambda'_{adv}$.

$\lambda_{dis}$ is only discriminator training, which is relatively trivial according to our discriminator descriptions in Appx. D.2.1.

### D.2.3    METRIC

Root mean square deviation (RMSE) is a commonly used metric, but it suffers difficulties when comparing datasets with different value scales. Therefore, we normalize it using its standard deviation.

$$\mathrm{NRMSE} = \frac{\sqrt{\mathbb{E}_{\mathbf{X}\sim p}\|X - \hat{X}\|_2^2}}{\mathrm{Std}(\mathbf{X})} \qquad (26)$$

---

[2]The code will be released upon acceptance.

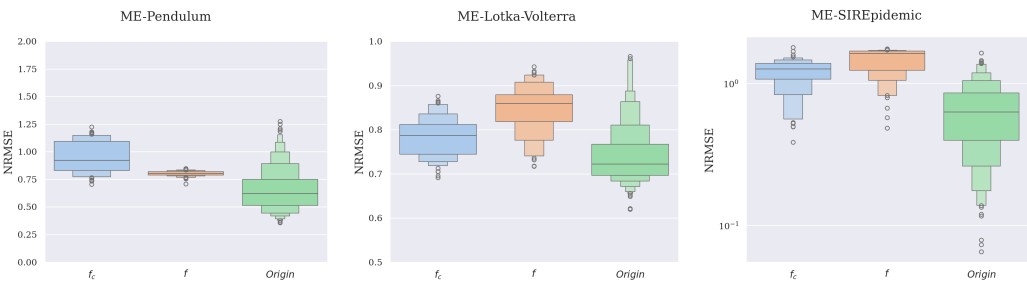

Figure 10: **Ablation study** on 3 DIF variants under 3 multi-environment ODE systems. For each pipeline, we provide model candidates with 50+ random hyper-parameter selections in their searching spaces, *i.e.*, more than 450 models in the figure.

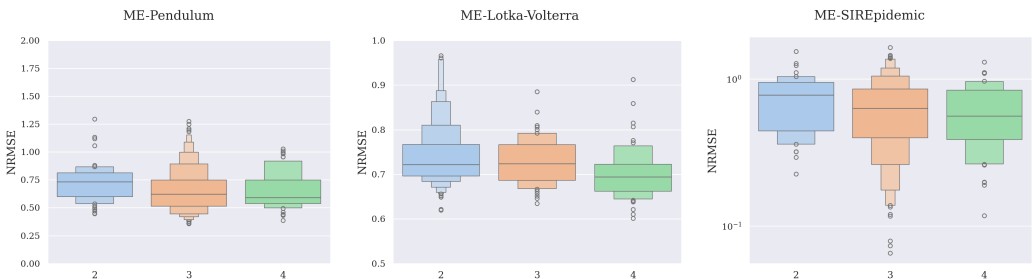

Figure 11: **Trajectory input length study** on models trained with different training input length factors under 3 multi-environment ODE systems.

### D.3 SOFTWARE AND HARDWARE

Our implementation is under the architecture of PyTorch Paszke et al. (2019). The deployment environments are Ubuntu 20.04 with 48 Intel(R) Xeon(R) Silver, 4214R CPU @ 2.40GHz, 755GB RAM, and graphics cards NVIDIA RTX 2080Ti.

## E SUPPLEMENTARY EXPERIMENTS

### E.1 ABLATION STUDY

As a complementary ablation study of Sec. 5.5, we train two models by eliminating two important components from the original model, namely, $f$ pipeline and $f_c$ pipeline. The $f$ pipeline removes the discriminator and only output $\hat{f}$ to the forecasting, which neglects the disentanglement process. The $f_c$ pipeline prunes the $\hat{f}$ output while maintaining the adversarial training process. As shown in Fig. 10, both $f$ and $f_c$ pipelines fail to perform valid invariant function learning aligning with our theoretical results. Specifically, the $f$ pipeline faces difficulties in extracting invariant functions without environment information. The unsatisfactory performance of the $f_c$ pipeline is attributed to the discriminator's training failure. This is because the training of the discriminator requires the capture of environment information, but the elimination of the $\hat{f}$ part also removes the training of $\hat{z}_e$, the critical environment information captor. Therefore, the discriminator loses the most important environment information input, leading to training failure.

### E.2 INPUT LENGTH AND ENVIRONMENT ANALYSIS

In order to ensure fairness, we fix the input length and the number of environments in our experiments. However, it is also interesting to figure out the effects of the input trajectory length $T_c$ and the number of environments on the model performance. Fig. 11 shows the performance of DIF given different input length factor $l_t$, where $T_c = \frac{T}{l_t}$. The results indicate the input length does not affect the performance significantly, where the only variances are attributed to the training difficulty of the

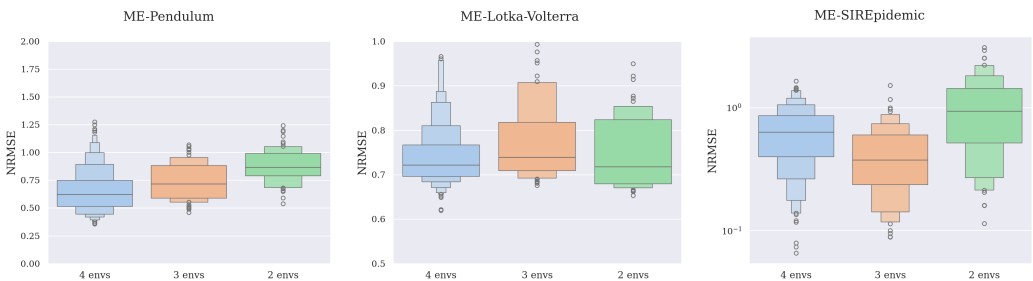

Figure 12: **Environment analysis** on models trained with different numbers of training environments under 3 multi-environment ODE systems.

Table 7: **Symbolic regression explanation comparisons.**

| Method | ME-Pendulum | | ME-Lotka-Volterra | | ME-SIREpidemic | |
|---|---|---|---|---|---|---|
| | NRMSE | SR Explanation | NRMSE | SR Explanation | NRMSE | SR Explanation |
| MAML | 0.9704 | $\frac{d\theta_t}{dt} = \omega_t - \frac{0.036\,(\theta_t + \omega_t)}{e^{\omega_t}}$ $\frac{d\omega_t}{dt} = \frac{\sin(\theta_t)}{-0.67} - 0.48\sin((\theta_t + 0.68)\sin(\omega_t))$ | 0.6774 | $\frac{dp_t}{dt} = q_t\frac{1}{p_t + 0.63}(-0.022)$ $\frac{dq_t}{dt} = \frac{p_t}{0.17} - q_t + q_t q_t$ | 0.2673 | $\frac{dS_t}{dt} = \frac{-S_t I_t - I_t + \cos(S_t)}{0.69}$ $\frac{dI_t}{dt} = 0.49 S_t + e^{\sin(0.45 S_t)} - 0.11$ $\frac{dR_t}{dt} = \cos(I_t \sin(S_t))(-0.087)$ |
| CoDA | 0.9695 | $\frac{d\theta_t}{dt} = \omega_t \rho\sin(\theta_t + \omega_t + 0.94) + \omega_t + 0.074$ $\frac{d\omega_t}{dt} = \frac{\sin(-\theta_t + \omega_t(-0.20) + 0.37)}{0.49} - 0.57$ | 0.7097 | $\frac{dp_t}{dt} = (1.1 - p_t p_t)\,0.54$ $\frac{dq_t}{dt} = (p_t + p_t)(p_t + q_t(-0.26)) - 0.66$ | 0.3184 | $\frac{dS_t}{dt} = \frac{-S_t + \sin(S_t \cdot 0.44)(-2.1)}{0.44}$ $\frac{dI_t}{dt} = \frac{3.5 S_t}{S_t + \frac{I_t}{S_t - 0.54}} + 0.097$ $\frac{dR_t}{dt} = \frac{S_t}{I_t + I_t + \frac{e^{S_t}}{I_t}} + 0.029$ |
| IRM | 0.7042 | $\frac{d\theta_t}{dt} = \omega_t \cdot 0.93$ $\frac{d\omega_t}{dt} = -\theta_t \alpha + \rho$ | 0.6989 | $\frac{dp_t}{dt} = -0.012$ $\frac{dq_t}{dt} = 0.083$ | 0.9768 | $\frac{dS_t}{dt} = e^{-S_t + \sin(S_t)} - 1.7$ $\frac{dI_t}{dt} = \sin\left(\frac{S_t}{S_t + S_t + \frac{\beta}{S_t}} - -0.12\right)$ $\frac{dR_t}{dt} = 0.33 - 0.021 S_t$ |
| VREx | 0.7274 | $\frac{d\theta_t}{dt} = \omega_t \cdot 0.92$ $\frac{d\omega_t}{dt} = \alpha(-1.1)\theta_t$ | 0.6877 | $\frac{dp_t}{dt} = -0.032$ $\frac{dq_t}{dt} = 0.12$ | 0.4652 | $\frac{dS_t}{dt} = S_t(-I_t - 0.10\beta)$ $\frac{dI_t}{dt} = \frac{S_t \beta}{S_t + S_t + \frac{\beta}{S_t}} + 0.078$ $\frac{dR_t}{dt} = 0.070 + \frac{\sin(\beta)}{e^{I_t}}$ |
| Ours | 0.3561 | $\frac{d\theta_t}{dt} = 0.99\omega_t$ $\frac{d\omega_t}{dt} = -0.97\alpha^2\sin(\theta_t)$ | 0.6194 | $\frac{dp_t}{dt} = 1.254 p_t - 0.38 q_t p_t$ $\frac{dq_t}{dt} = 4.1 p_t - 0.30 q_t - \gamma$ | 0.0652 | $\frac{dS_t}{dt} = -1.7 S_t I_t$ $\frac{dI_t}{dt} = 0.42 S_t I_t$ $\frac{dR_t}{dt} = -0.0088$ |
| GT | NAN | $\frac{d\theta_t}{dt} = \omega_t$ $\frac{d\omega_t}{dt} = -\alpha^2\sin(\theta_t)$ | NAN | $\frac{dp_t}{dt} = \alpha p_t - \beta p_t q_t$ $\frac{dq_t}{dt} = \delta p_t q_t - \gamma q_t$ | NAN | $\frac{dS_t}{dt} = -\beta\frac{S_t I_t}{S_t + I_t + R_t}$ $\frac{dI_t}{dt} = \beta\frac{S_t I_t}{S_t + I_t + R_t}$ $\frac{dR_t}{dt} = 0$ |

transformer given different input lengths. Therefore, a shorter input length can perform slightly better given the same training steps.

For the environment analysis, in addition to evaluations on the full set of environments, we benchmark model performance on datasets with three and two training environments. Specifically, we select [Powered, Air, Spring] and [Powered, Air] for ME-Pendulum; [Save, Fight, Resource] and [Save, Fight] for ME-Lotka-Volterra; and [Negative, Origin, Enlarge] and [Negative, Origin] for ME-SIREpidemic. While not all possible environment combinations are evaluated, these selections provide intriguing insights. As illustrated in Fig. 12, changes in the set of environments weakly affect model performance on ME-Lotka-Volterra. For ME-Pendulum, however, the inclusion of each additional environment consistently improves model performance. On ME-SIREpidemic, the performance boost observed with "3 envs" underscores the critical role of the environment Enlarge.

Two key observations regarding the ME-SIREpidemic are worth noting. First, the average performance degradation on "4 envs" suggests a reduced focus on the important environment Enlarge due to the addition of the final environment Loop. Second, the improvement in the best performance candidate demonstrates the additional benefits of the environment Loop. These findings illustrate that while adding environments can enhance the best possible discovery of invariant functions, it also increases the average training complexity that may cause average performance degradations.

### E.3 SYMBOLIC REGRESSION EXPLANATION COMPARISONS

To further evaluate the performance differences between the proposed method and the baselines, we apply PySR to the four baseline methods and obtain analytical explanations, as summarized in Tab. 7. The symbolic regression results provide an intuitive understanding of performance in relation to different NRMSE values. Specifically, when the NRMSE approaches 1, the resulting explanations are largely meaningless. As the NRMSE decreases to around 0.7, the explanations become more interpretable but may sometimes converge to oversimplified expressions, such as the IRM and VREx results on ME-Lotka-Volterra. When the NRMSE approaches zero, the expressions become more reasonable but are not always ideal due to the inherent limitations of PySR. This suggests that strong model performance does not necessarily guarantee high-quality explanations, highlighting the performance constraints of the explainer (PySR).

## F EFFICIENT HYPERNETWORK IMPLEMENTATION

One of the major challenges that limits the usage of hypernetworks is the implementation complexity. Most current implementations requires either re-implementing basic neural networks (von Oswald et al., 2020) or assigning predicted weights to the main function (forecaster) one by one for each forward pass (Ortiz et al., 2023; Sudhakaran, 2022; Kirchmeyer et al., 2022). To overcome these issues, we propose a *Reference-based* hypernetwork implementation technique that uses pure PyTorch without introducing any new modules or CUDA kernels. Our proposed technique does not require reassigning weights for each sample in one forward pass, *i.e.*, for any continuous $N$ training iterations with batch size of $B$ and a forecaster with $M$ parameter variables, our computation complexity is $O(NM + BM)$, instead of $O(BMN)$ as previous implementations.

Specifically, for every forward pass, we create a function parameter vector buffer $\in \mathbb{R}^m$ with fixed storage space, instead of reshaping and assigning the predicted function parameters with complexity $O(BM)$. As shown in Fig. 13, we consider the derivative neural network parameter variables as storage space pointers, *i.e.*, the network stores references instead of matrices. The fractions of function parameter vector buffer are pointed by these pointers; thus, once the buffer's values change by the predicted function parameters, *e.g.*, $\hat{f}$, the derivative network's parameters will be changed automatically without any assignment operators. To maintain the buffer's fixed storage space, several in-place operations are applied to maintains computational graphs and gradients.

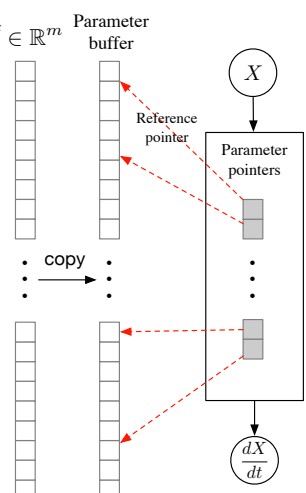

Figure 13: **Reference-based hypernetwork implementation.**

### F.1 EFFICIENCY COMPARISONS

Table 8: **Hypernetwork implementation efficiency comparisons**

| Implementation | Vectorization | Copy | Reference | First Step Time (s) | Avg Time $\pm$ Std (s) | Speedup |
|---|---|---|---|---|---|---|
| Non-vectorized | ✗ | ✓ | ✗ | 0.2466 | $0.1818 \pm 0.0601$ | 1x |
| Module-based | ✓ | ✓ | ✗ | 0.1768 | $0.1513 \pm 0.0737$ | 1.2x |
| Functional-based | ✓ | ✗ | ✗ | 0.2013 | $0.0198 \pm 0.0007$ | 9.2x |
| Ours | ✓ | ✗ | ✓ | 0.1805 | $0.0108 \pm 0.0006$ | 16.8x |

To evaluate the efficiency of our hypernetwork implementation, we compare it against several common implementation approaches. Specifically, we measure the forward pass time of our model over 200 continuous iterations in training mode, recording both the time for the first iteration and the average time for the subsequent iterations. While the first iteration typically takes a similar amount of time across all implementations, their performance diverges significantly in the subsequent iterations. As shown in Tab. 8, the *Non-vectorized* implementation represents methods that do not vectorize the

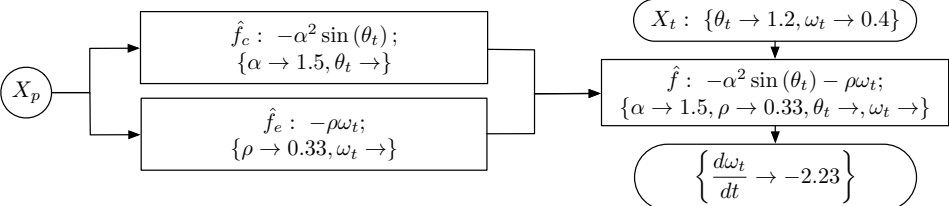

Figure 14: A function learning example.

derivative function and therefore must run different derivative functions sequentially. Approaches like CoDA (Kirchmeyer et al., 2022) attempt to vectorize the model by employing group-based convolution networks. However, these module-based implementations rely on stateful PyTorch modules, requiring the derivative function module to be replicated during each forward pass, which slows down the process. While these *Module-based* implementations offer a slight improvement over non-vectorized methods due to the vectorization benefits, the performance gain is limited.

In contrast, our vectorized *Functional-based* implementation leverages PyTorch's functional methods, achieving 9.2x speedup by avoiding the overhead associated with stateful modules. Note that vectorizing hypernetworks using libraries such as hypnettorch (von Oswald et al., 2020) can deliver similar speedups. Finally, our *Reference-based* implementation, which eliminates parameter assignment after the first iteration, nearly doubles the forward pass speed (16.8x) compared to implementations that require such assignments. Notably, this optimization remains applicable for potential future CUDA-based parallel hypernetwork implementations.

## G    SUPPLIMENTARY EXPLANATIONS

The diagram 14 illustrates the conceptual function learning process associated with the DIF framework, exemplified by the prediction of $\frac{d\omega}{dt}$. Each (hidden) function representation, such as $\hat{z}_c$, $\hat{z}_e$, $\hat{f}_c$, and $\hat{f}_e$, *implicitly* encodes both a symbolic function form and a value-assignment/substitution rule. For instance, $\hat{f}_c$ represents two types of information: (1) the function form, e.g., $-\alpha^2\sin(\theta_t)$, and (2) a partial assignment of values, such as $\{\alpha \to 1.5, \theta_t \to\}$, where $\alpha$ is assigned a value of 1.5 while $\theta_t$ remains a symbolic variable. This implies that $\hat{f}_c$ represents the function $-1.5^2\sin(\theta_t)$, which can accept input values in the form $\{\theta_t \to?\}$, effectively evaluated as $\hat{f}_c(?)$. Similarly, $\hat{f}$ is a function defined by two symbolic variables and accepts inputs in the form $\{\theta_t \to?, \omega_t \to?\}$. In this specific example, the input to $\hat{f}$ is $\{\theta_t \to 1.2, \omega_t \to 0.4\}$, resulting in the evaluation $\hat{f}(1.2, 0.4) = -2.23$.

## H    VISUALIZATIONS ON ME-LOTKA-VOLTERRA AND ME-SIREPIDEMIC

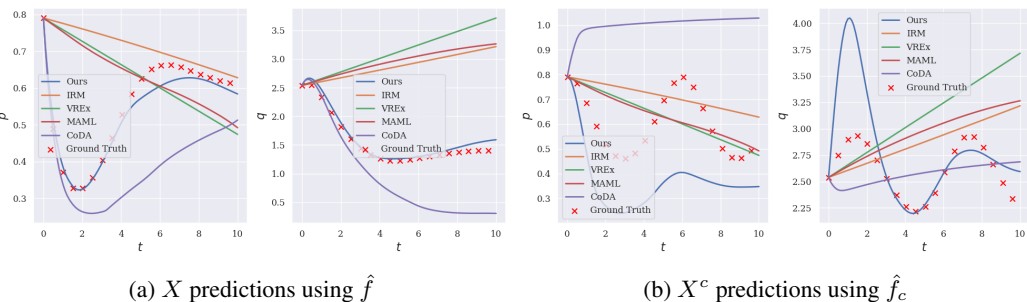

(a) $X$ predictions using $\hat{f}$        (b) $X^c$ predictions using $\hat{f}_c$

Figure 15: **Visualization of trajectory predictions on ME-Lotka-Volterra**

In this section, we present visualization comparisons for ME-Lotka-Volterra (Fig. 15) and ME-SIREpidemic (Fig. 16). The results for ME-SIREpidemic closely align with its quantitative findings. For the more challenging task of ME-Lotka-Volterra, our method's predicted trajectories remain closer to the ground truth. In the $X^c$ predictions, where most methods fail, our predicted trajectory has turning points closest to the ground truth in terms of timing, although there are deviations in magnitude (Fig. 15b). The complexity of the ME-Lotka-Volterra task arises from several factors, including the introduction of exponential functions within environments, the distribution of environments, and

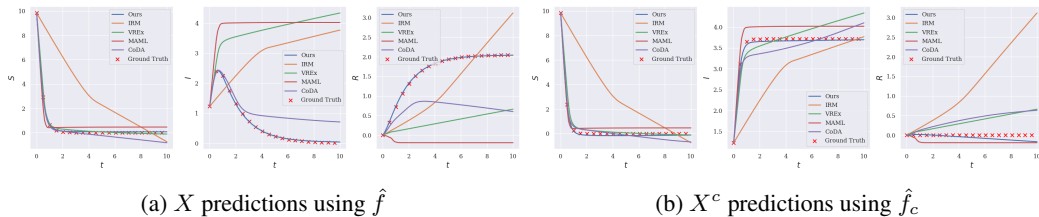

(a) $X$ predictions using $\hat{f}$        (b) $X^c$ predictions using $\hat{f}_c$

Figure 16: **Visualization of trajectory predictions on ME-SIREpidemic**

the limited number of samples in each environment. Addressing these challenges requires carefully designed benchmarks by domain experts, which we will discuss further in the limitations section.

# I LIMITATIONS AND FUTURE WORK

We discuss the limitations and the cover scope of this work; thus, shedding light on potential future directions accordingly.

First, in this invariant function learning tasks, we only target the function combinations like $f(x)+g(x)$ and $f(x)(1+g(x)) = f(x)+f(x)g(x)$ where the invariant and the environment functions are entangled by additions. We ignore the scenarios that the invariant and environment functions are entangled in forms like multiplications. Therefore, it may be of interest to disentangle more complex scenarios. However, for physical systems, given proper derivatives, factors are generally combined by additions, making the addition situations we considered important and widely applicable.

Second, in this work, we only consider invariant function learning on ODE systems; thus, it will be of interest to explore invariant function in PDE systems.

Third, although we prove the invariant function learning principle using information theory, it would still be quite significant to further build invariant error bounds for this invariant function learning task using VC-Dimension Vapnik & Chervonenkis (2015) or PAC-Bayesian McAllester (1998); Catoni (2007), which would provide more in-depth insights.

Forth, in this work, we propose 4 multi-environment ODE systems, but we are still lacking comprehensive benchmarks that built by domain experts. There are many questions remaining, such as how many environments are sufficient to infer invariant function, what are the requirements of the environment distributions to make it possible for invariant function learning.

Last but not least, unlike meta-parameters, the learned invariant functions have broader adaptation capabilities. For instance, a learned physical law can generalize to any system. It would be fascinating to explore applications of invariant function learning, such as learning generalizable physics from videos or developing physics-aware agents. Additionally, it would be interesting to investigate whether invariant function learning can contribute to the development of foundational models in physics.

