# OpenReview forum: "Discovering Physics Laws of Dynamical Systems via Invariant Function Learning"
_ICLR.cc/2025/Conference — Submitted to ICLR 2025_

### Official Review · Reviewer_mFCF · 2024-10-17

**Soundness:** 3
**Presentation:** 1
**Contribution:** 3
**Rating:** 6
**Confidence:** 3

**Summary:**

This paper considers the problem of learning invariant mechanisms that are shared across similar dynamical systems governed by ODEs observed in multiple environments. It introduces a new method, Disentanglement of Invariant Functions, to tackle this problem using a hypernetwork for function prediction and then a separate approach for forecasting.

The numerical examples show that invariant learning techniques outperform meta-learning methods for that task, and that the proposed approach performs the best among the invariant learning techniques compared. In these examples, the invariant functions extracted are more or less aligned with the true invariant mechanisms. A new hypernetwork implementation is presented and shown to significantly improve efficiency.

**Strengths:**

**Problems Tackled**

- Proposes a new approach for a problem that has perhaps not been tackled enough

- Considers harder problems than in the existing literature (the environments considered cover changes beyond function coefficients and extend to entirely different functional forms)

**Results**
- The proposed approach outperforms the other existing approaches used as baselines

- Proposes a new hypernetwork implementation that can significantly improve efficiency

**Weaknesses:**

**Clarity**
The proposed approach is not presented clearly. The details of the various components are scattered in different parts of the paper and its appendix and sometimes only implicitly. The reader has to piece it out together himself and sometimes guess how certain components are designed and it is easy to misunderstand the proposed approach. Here are a few examples, comments, and suggestions:

- The two-phase process should be highlighted more explicitly than hidden in a few paragraphs. Adding structure with bold keywords, or lists, or bullet points could help. Same comment about the two main challenges faced, and then later how they are addressed.

- The details of how the forecasting phase is performed should be made more explicit. Line 164 suggests that a numerical integrator is used for that part and Line 193 suggests instead that the forecasting part is done using neural networks. The end of Section 3.2.1 suggests that the numerical integrator might be replaced during training, and the end of Section 5.2 mentions that a numerical integrator is used for the invariant dynamics. This needs to be clarified further, by stating explicitly what the different contexts are and what is used for forecasting in each context.

- The paragraph around lines ~134-143 and Table 1 are not clear enough. "defining environments as discrete interventions (Pearl, 2009) and model them as a categorical environment variable" needs to be explained in more details, and these keywords should be defined for readers that are not familiar with the cited literature. What does distribution represent in the table? Need more examples of different environments for "our environment" to better grasp how much variability there is in the dataset. Using colors in the table instead of underlining could also improve readability.

- It could be useful to have a higher-level version of the upper part of Figure 2 earlier in the paper to explain the two phases ion more details.

- The lower part of Figure 2 is unclear. Why only partial assignments? What process does each of the arrows represent? I don't understand the "flow" of that figure. Are the outputs of the hypernetwork exactly in this symbolic form, or is this after applying some form of symbolic regression that is not part of the proposed approach? In the same direction, the sentence around line 163, "Intuitively, taking Fig. 1a as an example, this phase aims at the reasoning of the function basis sin(θt), −ωt, ωt/|ωt| , and the coefficients α, ρ" is too imprecise. It is unclear what the output of the hypernetwork is. The quoted sentence and the figures seem to insinuate that a symbolic function form is learnt explicitly and then used as the differential equation to solve in the forecasting phase. However, later on, in section 5.6, symbolic regression using PySR is applied to the output of the hypernetwork, which shows that a symbolic function form was not learnt explicitly. This needs to be clarified, it should be written very explicitly what the output of the hypernetwork looks like and how it is taken into account in the forecasting phase.

- I would recommend adding more details about the new hypernetwork implementation in the main text instead of the appendix, given that it could be an important contribution of the paper based on the speedups observed. This also depends on whether other similarly or more efficient hypernetwork implementations have been proposed in other works and are not mentioned here (in which case they should)

- There are also numerous grammatical and spelling errors, especially in Section 3 and Appendix B (10+ in each of them)

- Limitations and Future Work should not be relegated to Appendix G and should appear in the main text




**Results**

- Aside from the pendulum, the symbolic regression explanations are quite off from the ground truth, suggesting a certain lack of generalization capability (although it should be noted that these examples are harder than those considered in previous literature)

**Questions:**

Questions and suggestions were outlined made in the Weaknesses section.

Overall, I do believe this paper tackles an important problem and makes some contribution towards solving that problem, but the clarity and presentation of the paper are not good enough.

I would be happy to revise my review if the authors significantly improve the clarity and readability of the paper

---

> ### Author Response · Authors · 2024-11-21
> **Official comments (Part 1)**
>
> Dear Reviewer mFCF,
>
> We sincerely thank you for your detailed feedback and thoughtful suggestions. Your comments have been instrumental in helping us identify areas for improvement, and we are deeply grateful for the time and effort you have invested in reviewing our work. Based on your feedback, we have made substantial revisions to significantly improve the clarity, presentation, and overall quality of the paper. Below, we address each of your points individually and describe the changes made in response.
>
> > The two-phase process and two main challenges should be highlighted
>
> Thank you for this suggestion. As you recommended, we have explicitly highlighted the two-phase process and the two main challenges in the main text. To improve readability, we used bolded keywords and lists to clearly structure these discussions, ensuring they are easy to locate and understand.
>
> > The details of how the forecasting phase is performed should be made more explicit
>
> We appreciate your emphasis on this point. To address it, we have added a dedicated paragraph at the end of Section 3.1 that directly explains the forecasting process. Specifically, we clarified that the numerical integrator requires a customized derivative function, which is represented as a neural network with parameters $\hat{f}$ predicted by the hypernetwork. For simplicity, the notation $\hat{f}(\cdot)$ is used to denote the neural network function parameterized by $\hat{f}$.
>
> > The paragraph around lines ~134-143 and Table 1 are not clear enough.
>
> Thank you for pointing out this area of confusion. We have made several efforts to improve this section:
> 	1.	We rewrote the paragraph under the title “Function Environments,” providing a detailed explanation of the differences between Function Environments (formerly “our environments”) and Coefficient Environments (formerly “micro-environments”).
> 	2.	Unnecessary descriptions were removed to enhance clarity and avoid confusion.
> 	3.	Table 1 was improved with clearer formatting, and we replaced underlining with colors to enhance readability.
>
> We hope these revisions make the concepts and examples more intuitive for readers.
>
> > It could be useful to have a higher-level version of the upper part of Figure 2 earlier in the paper to explain the two phases ion more details.
>
> Thank you for this excellent suggestion. We reformulated the general causal graph into a Structural Causal Model (SCM) in Figure 2, explicitly splitting it into two phases to provide a high-level understanding. This higher-level version is introduced earlier in the paper to align with your feedback and facilitate reader comprehension.
>
> > The lower part of Figure 2 is unclear.
>
> We appreciate your observation regarding the lower part of Figure 2. This section was intended to provide an intuitive explanation of the process but, as you pointed out, it did not fully reflect the precise details of the DIF framework. Specifically:
> 	1.	All representations in the DIF framework are hidden vectors, meaning there is no explicit symbolic form or symbolic regression in the framework.
> 	2.	Partial assignments indicate that the output remains a function rather than a specific value after assignments.
>
> To address this, we have moved the lower part of Figure 2 to the appendix, where it can serve as a supplementary intuitive explanation. In its place, we provide a more precise description of the DIF framework in the main text to avoid ambiguity.
>
> > I would recommend adding more details about the new hypernetwork implementation in the main text instead of the appendix, given that it could be an important contribution of the paper based on the speedups observed.
>
> We truly appreciate your suggestion to emphasize the hypernetwork implementation. In response, we have expanded the discussion of the hypernetwork in the main text, while still referring readers to the appendix for additional technical details. This balance was necessary to accommodate space limitations while highlighting the speedups observed and the contribution of this implementation.
>
> > There are also numerous grammatical and spelling errors, especially in Section 3 and Appendix B (10+ in each of them)
>
> Thank you for bringing this to our attention. We have reviewed the paper and corrected all grammatical and spelling errors, especially in Sections 3 and Appendix B. We hope the revised version is significantly more polished and professional.

---

> ### Author Response · Authors · 2024-11-21
> **Official comments (Part 2)**
>
> > Limitations and Future Work should not be relegated to Appendix G and should appear in the main text
>
> You are absolutely right. We have moved the discussion of limitations and future work into the main text, as this is crucial for presenting a balanced and transparent narrative. Thank you for suggesting this improvement.
>
> > Aside from the pendulum, the symbolic regression explanations are quite off from the ground truth, suggesting a certain lack of generalization capability
>
> Thank you for pointing this out. We would like to clarify that symbolic regression (SR) is used for explanations and does not fully reflect the generalization capability of the neural network. Specifically:
>
> - **Good PySR results indicate good generalization ability, but good generalization ability does not guarantee good PySR results**. PySR faces challenges when dealing with datasets containing many variables, leading to an extremely large search space and often suboptimal results.
> - NRMSE on the invariant set provides a more reliable measure of generalization capability. For instance:
>   - In the ME-SIR Epidemic dataset, a near-perfect result is reflected by an NRMSE of 0.0652.
>   - In the ME-Lotka-Volterra dataset, the invariant function learning result achieves an NRMSE of 0.6194. While suboptimal, it is still decent compared to the in-distribution test result of 0.3881 NRMSE (a lower bound), especially given the complexity present in the dataset
>
>
> Additionally, **Appendix E.3** now includes a detailed comparison of symbolic regression explanation results, which further demonstrates the strength of our approach compared to baselines.
>
> --------------
>
> We sincerely hope that these revisions address your concerns and significantly improve the clarity and readability of our work. Thank you once again for your thoughtful review and valuable suggestions—they have played a crucial role in enhancing the quality of this paper!
>
> Best,
>
> Authors

---

> > ### Author Response · Authors · 2024-11-25
> >
> > Dear Reviewer mFCF,
> >
> > We hope this message finds you well!
> >
> > We are writing to kindly remind you to review our revised submission, which **incorporates more intuitive explanations and precise figures to enhance our presentation based on your constructive suggestions.** We would greatly appreciate it if you could let us know whether our dedicated revision has effectively addressed your major concerns.
> >
> > Thank you once again for the time and effort you have devoted to providing thoughtful reviews of our work. If there are any clarifications or additional information we can provide to assist you further, please do not hesitate to let us know.
> >
> > Sincerely,
> >
> > Authors

---

> > > ### Comment · Reviewer_mFCF · 2024-11-26
> > >
> > > I would like to thank the authors for their response to my review. **The paper is significantly clearer in my opinion, and the quality of the paper has also improved. **
> > >
> > > **I am updating my rating from a 5 to a 6, and would have selected 7 if it had been an option.**

---

> > > > ### Author Response · Authors · 2024-11-26
> > > >
> > > > Dear Reviewer mFCF,
> > > >
> > > > We are delighted to hear that you find the revised version significantly clearer and of improved quality. Thank you again for your valuable feedback and support in enhancing our work.
> > > >
> > > > Sincerely,
> > > >
> > > > Authors

---

### Official Review · Reviewer_Fxop · 2024-10-29

**Soundness:** 2
**Presentation:** 2
**Contribution:** 2
**Rating:** 3
**Confidence:** 3

**Summary:**

The paper proposes an approach to predict the dynamics functions of ordinary differential equation-based dynamical systems. The particular twist in this paper is that all observations are assumed to be governed by unique dynamics functions that are only partially shared across observations. The goal is to infer the common part of the dynamics function that is shared across all observations. The difficulty is that this shared part is not observed directly in any observation.

**Strengths:**

The paper proposes a novel problem setting that has not been tackled so far. The writing is for the most part very clear (with the exception of the theoretical section, see my comments below) and the figures greatly illustrate the idea of the paper. The authors propose an empirically evaluate a new computational model and also attempt to theoretically motivate their approach (see comments below though). Given the novel and innovative problem setting, there seem to not be any readily-applicable baseline models out there. The authors therefore also propose / adapt several baseline models to compare their approach to - and show empirically that the proposed approach outperforms these baseline models.

**Weaknesses:**

My main issue with this paper concerns the theoretical section - there are some things that I either do not understand / find hard to follow or that are incorrect.

Theorem 3.1 states that the predicted function is equivalent to the true function if and only if the invariant function learning principle of equation 2 is satisfied. However, equation 2 is not a a equation (or inequality) - it rather boils down to a real number (the mutual information between predicted and true function). I don’t really understand what it is supposed to mean to satisfy a real number.

Moreover, equation 2 compares the true function and the predicted function in terms of (conditional) mutual information. One thing that is unclear to me though is what the distribution of the true function f is supposed to be: to my mind, there is presumably exactly one true underlying function and hence the distribution is a dirac delta impulse. However, the mutual information between a dirac delta and any different function seems to be always zero, which makes me question this statement. (You can find a derivation for discrete random variables here, I believe the same derivation also works for continuous random variables: https://stats.stackexchange.com/questions/25095/mutual-information-with-a-dirac-delta-type-pdf).

In section 3.2.1 you train encode and decoder by  “apply[ing] the cross-entropy minimization.” According to the following lemma 3.2, however, it seems that you instead use the MSE as objective function. What is the reason to first say that you use CE while in practice (and theory) you actually use the MSE?

Lemma 3.2 states that under a Gaussian relaxation (which is not motivated in any way), optimizing the cross-entropy is equivalent to optimizing the MSE; this could be made more precise since what the authors mean is probably that the optimal minimizer is the same. (The actual objective values at their respective minimum, however, are most likely not the same).

(Minor:) In equation 4a, I suggest to replace $\hat{X}^c$ with $h_{\theta_c}(X_p)$ so that it is immediately clear where parameters $\theta_c$ enter the objective. Moreover, I find the statement “where $\hat{X}^c$ is the trajectory predicted by $P(X|h_{\theta_c}(X_p), X_0)$” slightly confusing since P is a distribution whereas X is a prediction (/estimate). If I understand correctly, $\hat{X}^c$ is simply the output of $h_{\theta_c}(X_p)$, integrated using a numerical solver with initial value $X_0$. (In other words, simply saying that $\hat{X}^c$ is predicted by the distribution P leaves open the question of how exactly X was obtained from the distribution.)

These issues are the main reason for the currently assigned presentation and soundness (as well as overall) scores.

**Questions:**

- The symbolic regression literature has proposed several works using transformers for dynamical systems inference [1,2,3], which I believe may also be regarded as hyper networks. I think it would be good to include a statement in the related work section on how they relate to your work.

- An interesting additional baseline to your problem could be to run PySR on the different environments directly. Since PySR predicts symbolic function, one might naively think that this should be sufficient to discover the invariant part of the equations that is shared across all systems. I.e. PySR would spit out a environment-specific equation for each environment but these might already contain the shared parts so that no additional modeling that explicitly takes this problem setting into account would be necessary. It would be good to see results on this to judge the value of your approach.

- In section 2.2.1 you mention (at the end of page 3) that “This reverse inference requires causal mapping f → X to be injective.” How reasonable is this assumption / requirement? It seems to me that this is in general (i.e. without restrictions on the function class that f comes from) is violated, since there can be multiple functions that generate the same output (especially if the output is from a finite interval, as it will be in practice.)

- Where do you see applications of your approach in practice? Have you tried evaluating it on any real world dataset that contains noise, missing data etc.?

References:
[1] Becker et al., (2023) "Predicting Ordinary Differential Equations with Transformers"
[2] d'Ascoli et al., (2024) "Odeformer: Symbolic regression of dynamical systems with transformers"
[3] Seifner et al., (2024) "Foundational Inference Models for Dynamical Systems"

---

> ### Author Response · Authors · 2024-11-21
> **Official comments (Part 1)**
>
> Dear Reviewer Fxop,
>
> We sincerely thank you for your insightful and detailed feedback, which has greatly helped us improve both the clarity and rigor of our work. Your comments reflect a deep understanding of the topic, and we deeply appreciate the opportunity to address your concerns and provide additional context. Below, we provide responses to each of your valuable points.
>
> > What does the invariant function learning principle satisfied mean?
>
> Thank you for pointing this out. To improve clarity, we have reformulated the invariant function learning principle (an optimization principle) as follows:
>
> Theorem 3.1 (Invariant function learning principle). Given the causal graph in Fig. 2, and the predicted function random variable $\hat{\mathrm{f}}\_c=h\_{\theta\_c}\left(\mathbf{X}\_p\right)$, it follows that the true invariant function random variable $\mathrm{f}\_c=h\_{\theta\_c^*}\left(\mathbf{X}\_p\right)$, where $\theta_c^*$ is the solution of the following optimization process, described as
>
> $$
> \theta\_c^*=\underset{\theta\_c}{\arg \max } I\left(h\_{\theta_c}\left(\mathbf{X}\_p\right) ; \mathbf{f} \mid \mathbf{X}\_0\right) \text { s.t. } h\_{\theta\_c}\left(\mathbf{X}\_p\right) \perp \\! \\! \perp \mathbf{e},
> $$
>
> where $I(\cdot ; \cdot)$ is mutual information that measures the information overlap between the predicted invariant function random variable $h_{\theta_c}\left(\mathbf{X}_p\right)$ and the true full-dynamics function random variable f.
>
> > Does the $\mathbf{f}$ in $I\left(h_{\theta_c}\left(\mathbf{X}_p\right) ; \mathbf{f} \mid \mathbf{X}_0\right)$ follow a Dirac delta distribution? What is the distribution of the true function f?
>
> This is an excellent question. To clarify, $\mathbf{f}$ is defined as a function random variable on a manifold rather than as a deterministic Dirac delta distribution. For example, consider the set of functions $\alpha^2 \sin \left(\theta_t\right)$, where $\alpha$ varies continuously. In this case, the distribution of $\mathbf{f}$ is determined by the continuous distribution of $\alpha$. Therefore, $H(\mathrm{f}) \neq 0$, and f is not deterministic. For further clarification, we have updated Section 2.2 and Table 1 to better illustrate the distinction between $f_1$ (a single function) and $\mathbf{f}_1$ (a function random variable).
>
> > Is $I\left(h_{\theta_c}\left(\mathbf{X}_p\right) ; \mathbf{f} \mid \mathbf{X}_0\right)$ always zero?
>
> Thank you for this thoughtful question. It is not zero. The original confusion might stem from interpreting $\mathbf{f}$ as a Dirac delta function, which would lead to $H(\mathbf{f})=0$. However, as clarified earlier, $H(\mathbf{f}) \neq 0$, and thus $I\left(h_{\theta_e}\left(\mathbf{X}_p\right) ; \mathbf{f} \mid \mathbf{X}_0\right)$ is non-zero. Additionally, Table 1 has been updated to emphasize that $\mathbf{f} \neq f$, and $\mathbf{f}_c \neq f_c$. This distinction is central to our contribution and novelty, as we allow for distributions of functions rather than treating functions as deterministic entities.
>
> > What is the reason to first say that you use CE while in practice (and theory) you actually use the MSE?
>
> Thank you for pointing out this discrepancy. MSE is used as the empirical loss, but we ensure that it aligns with our optimization goal based on information theory. Instead of taking MSE as a given, Lemma 3.2 and Proposition 3.3 bridge probability modeling with practical error measures. We hope this justification aligns with your expectations and improves clarity.

---

> ### Author Response · Authors · 2024-11-21
> **Official comments (Part 2)**
>
> > Why Gaussian relaxation?
>
> Thank you for highlighting this. To avoid unnecessary reliance on Dirac delta functions, we start from the forecasting modeling and introduce Gaussian noise modeling for greater realism. Specifically:
>
> ​1. We incorporated noise into our causal model, transitioning to a Structural Causal Model (SCM) as detailed in **Appendix B.1**.
>
> ​2. The forecasting process was reformulated to connect probability modeling with predictions, following probability inference techniques. This formulation enables us to model predictions probabilistically using Gaussian noise.
>
> For clarity, we added a detailed description of this process at the end of Section 3.1 and also post it here. We sincerely hope this addition clarifies the rationale behind our approach:
>
> Given the produced neural network function $\hat{f}$, we apply a numerical integrator as our forecastor, a function $g\_{\text {int }}$ that takes a derivative function $\hat{f}$ and initial states $\mathbf{X}\_0$ as inputs, to obtain $\hat{X}=g\_{\text {int }}\left(\hat{f}, X\_0\right)+\epsilon$ where $\epsilon$ is sampled from a Gaussian noise $\mathcal{N}\left(\mathbf{X} ; 0, \sigma^2 I\right)$ introduced by calculation deviations. This forecasting formulation enables the following probability modeling, where we obtain the forecasting given realizations $X_0$ and $\hat{f}$ as a Gaussian distribution $\mathcal{N}\left(\mathbf{X} ; g_{\text {int }}\left(\hat{f}, X_0\right), \sigma^2 I\right)$ denoted as $p\left(\mathbf{X} \mid \hat{f}, X_0\right)$. Therefore, in probability modeling, $\hat{X}$ is sampled from $p\left(\mathbf{X} \mid \hat{f}, X_0\right)$.
>
> This modeling is utilized in our Lemma to establish a connection between empirical optimization and probabilistic inference. Specifically, it demonstrates that minimizing the MSE effectively approximates minimizing an information-theoretic objective when small Gaussian noise is introduced. The inclusion of Gaussian noise, as proven, does not alter the optimization goal, thereby serving as a practical and theoretically grounded bridge.
>
> > How is $\hat{X}^c$ predicted by $P\left(X \mid h_{\theta_c}\left(X_p\right), X_0\right)$?
>
> As detailed in the Gaussian relaxation answer, $\hat{X}$ is sampled from $p\left(\mathbf{X} \mid \hat{f}, X\_0\right)=\mathcal{N}\left(\mathbf{X} ; g\_{\text {int }}\left(\hat{f}, X\_0\right), \sigma^2 I\right)$.  Similarly, $\hat{X}\_c$ is sampled from $p\left(\mathbf{X} \mid \hat{f}\_c, X\_0\right)$. Basically, it is sampled from $\mathcal{N}\left(\mathbf{X} ; g\_{\text {int }}\left(\hat{f}\_c, X_0\right), \sigma^2 I\right)$, i.e., $\hat{X}\_c = g\_{\text {int }}\left(\hat{f}\_c, X_0\right) + \epsilon$ where $\epsilon \sim \mathcal{N}\left(\mathbf{X} ; 0, \sigma^2 I\right)$.
>
> ----------
> Questions:
>
> > Transformer-based symbolic regression literature discussion.
>
> We are grateful for the references you provided. These works have now been incorporated into the related work section, where we discuss their relevance and distinctions in the context of our approach.
>
> > PySR as a baseline for invariant function discovery.
>
> Thank you for this intriguing suggestion! While PySR is indeed a powerful tool, it requires explicit inputs for all variables (e.g., $\theta, \omega$, and $\alpha$ ). In our dataset, $\alpha$ is unknown and must be inferred, which makes PySR unsuitable for this task. Instead, we use PySR as an explainer for symbolic regression in our framework, highlighting the additional value of our approach.
>
> > $\mathbf{f} \rightarrow X$​ being injective can be too restrictive.
>
> Thank you for raising this important concern. We agree that this assumption was overly restrictive, and in response, we have eliminated it in our updated theoretical framework. Noise and complex scenarios are now modeled using an enhanced causal model with updated proofs. Please refer to the **“Upgrade of Theoretical Analysis” section in the revision summary** for details. We hope this improvement addresses your concern.
>
> > Where do you see applications of your approach in practice? Have you tried evaluating it on any real world dataset that contains noise, missing data etc.?
>
> We believe invariant function learning represents an exciting hindsight for modeling dynamical systems. While this paper focuses on ODEs, the concept is more general and theoretically extensible to PDEs. However, such adaptations present challenges (e.g., sampling high-dimensional variables) that warrant further research. We plan to explore these applications and real world datasets in future work in collaboration with domain experts.
>
> ------------------
>
> Once again, we really appreciate your invaluable feedback and the opportunity to improve our work. Your thoughtful comments have inspired many of the changes and additions we’ve made, and we hope our responses address your concerns comprehensively. Thank you for your detailed and insightful review!
>
> Best,
>
> Authors

---

> > ### Author Response · Authors · 2024-11-25
> >
> > Dear Reviewer Fxop,
> >
> > Hope you are doing well! We are writing to kindly remind you to check our revised submission, which incorporates **more intuitive explanations to clarify possible confusions with enhanced theoretical analyses and extensive experiments based on your constructive suggestions and comments.**
> >
> > Could kindly let us know whether our dedicated revision and detailed explanations have addressed your major concerns? If there are any clarifications or additional information we can provide, please do not hesitate to let us know.
> >
> > Thank you again for your thoughtful review and the time and effort you have already dedicated to evaluating our work.
> >
> > Sincerely,
> >
> > Authors

---

> ### Comment · Reviewer_Fxop · 2024-11-25
> **Thank you for your answers**
>
> Thank you for the detailed answers.
>
> > To clarify, $\mathbf{f}$ is defined as a function random variable on a manifold rather than as a deterministic Dirac delta distribution. For example, consider the set of functions $\alpha^2 \sin \left(\theta_t\right)$, where $\alpha$ varies continuously. In this case, the distribution of $\mathbf{f}$ is determined by the continuous distribution of $\alpha$.
>
> Thank you, I now better understand how you are defining $\mathbf{f}$ per environment. However, this makes me wonder how sensible the Gaussian relaxation really is: assume $\mathbf{f}$ is given by $\alpha^2 \sin \left(\theta_t\right)$ as in your example above, and $\alpha$ follows a uniform distribution with support $[0, 1] \cup [100, 101]$. It seems to me that modelling such a distribution with a Gaussian is not a good idea and sampling predictions will in most cases lead to an incorrectly predicted function.
>
> > Thank you for this intriguing suggestion! While PySR is indeed a powerful tool, it requires explicit inputs for all variables (e.g.,
> $\theta$, $\omega$ and $\alpha$). In our dataset, is unknown and must be inferred, which makes PySR unsuitable for this task.
>
>
> Looking at the first example in your table 2, you are simulating a 2-variable dynamical system ($\theta$ and $\omega$) which has one parameter ($\alpha$). PySR can be applied to each simulated observation separately (i.e., for every single observation obtained for a fixed $\alpha$). This will give you as many predictions as you have observations. My question was in how far these predictions already agree with each other, i.e., would this not potentially already suffice to see the shared structure in the ground truth functions across (and also within) environments?

---

> > ### Author Response · Authors · 2024-11-25
> > **Clarifications**
> >
> > Thanks a lot for the reviewer's reply! We would like to provide further explanations and experimental results to address your concerns.
> >
> > > Using Gaussian is not a good idea to model the data distribution like uniform $\alpha$ distribution.
> >
> > Thank you for letting us know your understanding! We hope to clarify that:
> >
> > 1. The distribution of $\alpha$, $p(\alpha)$ is not constrained to be any specific form.
> > 2. The Gaussian distribution relaxation is a probability modeling for neural networks. Please allow us to use a simplified example to explain this concept. Given a neural network $f_{NN}(x)=x+1$, a function with parameters, we generally assume that $f_{NN}(\cdot)$ is deterministic, such as $f_{NN}(1)=1+1=2$. However, there are noises during the calculations, causing $f_{NN}(1)=1+1 +0.0000001=2.0000001$. Therefore, we model the $f_{NN}$ as $f_{NN}= x+1+\epsilon$, where $\epsilon$ is a very small Gaussian noise. In this paper, we inject the noise into the integration calculation process.
> >
> > Generally, we don't need to consider these noises. We explicitly consider these noises for just probability modeling that bridges the deterministic functions with probability distributions. The relaxation itself is trivial, but its corresponding probability modeling is helpful. It is not necessary to include this explicit probability modeling and use it as granted, but this work hopes to explain these details to readers.
> >
> > In conclusion, the Gaussian distribution relaxation is a bridge for theoretical analysis, **not related to the data prior distributions where we do not apply any Gaussian constraints.**
> >
> > > How far do these predictions already agree with each other, i.e., would this not potentially already suffice to see the shared structure in the ground truth functions across (and also within) environments?
> >
> > Thank you for the clarifications! We post the individual prediction results from several observations, which shows the agreement the reviewer is curious about.
> >
> > 1.
> > $$\begin{aligned}
> >         \frac{d\theta_t}{dt} &= \omega_{t} + \left(0.0023 - \omega_{t}\right) \cos{\left(\theta_t \right)} 0.0027 \\\\
> >         \frac{d\omega_t}{dt} &= \sin{\left(\theta_t \right)} \left(-1.6\right) + \sin{\left(\sin{\left(\frac{\sin{\left(\omega_{t} \right)}}{0.42} \right)} \right)} 0.18
> >         \end{aligned}$$
> > 2.
> > $$\begin{aligned}
> >         \frac{d\theta_t}{dt} &= \omega_{t} + \frac{3.2 \cdot 10^{-5}}{\omega_{t} + \frac{0.68}{\theta_t}} \\\\
> >         \frac{d\omega_t}{dt} &= \left(- \theta_t + \omega_{t} \left(\omega_{t} - 0.33\right)\right) \left(\omega_{t} + 2.1\right)
> >         \end{aligned}$$
> > 3.
> > $$\begin{aligned}
> >         \frac{d\theta_t}{dt} &= \omega_{t} e^{\left(-0.00089\right) \left(-1.8\right) \frac{1}{1.3} \frac{1}{-0.32 - \frac{0.28}{\omega_{t}}}} \\\\
> >         \frac{d\omega_t}{dt} &= - \theta_t + \frac{\omega_{t}}{- \sin{\left(\omega_{t} \cdot 1.7 \right)} - 1.3}
> >         \end{aligned}$$
> >
> > In these cases, researchers have to do **exhausted manual analyses and still cannot achieve our model performance**.
> >
> > We hope these explanations can address your concerns. Thank you!
> >
> > Sincerely,
> >
> > Authors

---

> > > ### Author Response · Authors · 2024-12-01
> > > **Rebuttal follow-up**
> > >
> > > Dear Reviewer Fxop,
> > >
> > > Thank you for your detailed review! We have carefully addressed your feedback and made corresponding revisions to the manuscript. As the discussion period is drawing to a close, could you kindly confirm if our clarifications have adequately addressed your remaining concerns?
> > >
> > > Thank you again for your constructive reviews and suggestions. We wish you a wonderful holiday season.
> > >
> > > Sincerely,
> > >
> > > Authors

---

> > > > ### Comment · Reviewer_Fxop · 2024-12-03
> > > > **Thank you for your comments**
> > > >
> > > > > In these cases, researchers have to do exhausted manual analyses and still cannot achieve our model performance.
> > > >
> > > > Thank you for the detailed answer about PySR's performance. Indeed, PySR does not seem to easily solve the problem that you are tackling with your model.
> > > >
> > > > > However, there are noises during the calculations, causing $f_{NN}(1)=1+1 +0.0000001=2.0000001$ .
> > > >
> > > > Thank you for your answer - I find it difficult to follow the reasoning here. Numerical integrators for ODEs are usually deterministic - hence I do not see where this noise should enter the calculations as you describe. I understand that you mostly use the Gaussian relaxation as a tool for your theoretical analysis though.
> > > >
> > > >
> > > > There are a few more things that I noted while giving the paper another read.
> > > >
> > > > - It is not clear to me how the function composition $g_{comp}$ in Appendix B.1 is supposed to work. If $f_c$ and is only caused by $c$ and $f_e$ is only caused by $e$, is it even certain that you can combine them into a single function $f$? How could such a function $g_{comp}$ look like in practice? For instance, to stay with the examples of Table 2, couldn't $f_e$ be an environment-specific realization of the Lotka-Volterra function while $f_c$ is the common underlying function of the Pendulum? In essence, I do not really understand the generative process here.
> > > >
> > > > - I also do not fully understand the setup of the predictive model. In section 3.1 you write
> > > > > The function space $\mathcal{F}$ consists of all possible neural networks with $m$ parameters, and a function $f \in \mathcal{F}$ can be represented as a vector in $R^m$.
> > > >
> > > > However, I do not see how a point in $R^m$ uniquely identifies any possible neural network (NN) with $m$ parameters: e.g. assume $m = 6$, two possible neural networks are e.g. 1) an 4-layer NN with 2-1-1-2 parameters per layer or 2) a 3-layer NN with 2-2-2 parameters per layer. These seem to model very different functions yet both could map to any point in $R^m$.
> > > >
> > > > In addition I also share the concern from other reviewers that it may not be easy to group observations into environments (that is, to obtain environment labels.) This also touches on my original question about potential applications which you answere as
> > > > > We believe invariant function learning represents an exciting hindsight for modeling dynamical systems. While this paper focuses on ODEs, the concept is more general and theoretically extensible to PDEs. However, such adaptations present challenges (e.g., sampling high-dimensional variables) that warrant further research.
> > > >
> > > > These are nice ideas for model extensions but, from my perspective, do not describe potential applications where inference from multiple minimally different environments (which after all is the main concern of this manuscript) needs to be considered.
> > > >
> > > >
> > > > So, overall, I appreciate the authors' effort to clarify the manuscript yet I will maintain my score.

---

> ### Author Response · Authors · 2024-12-03
>
> Dear Reviewer Fxop,
>
> > Numerical integrators for ODEs are usually deterministic.
>
> Thank you for your comment. While numerical ODE integrators are deterministic in principle, in practice, small discrepancies arise due to finite precision arithmetic and external factors like hardware-level disturbances (e.g., cosmic rays causing bit flips). These tiny variations, while deterministic in origin, can be effectively modeled as small Gaussian noise $\epsilon \sim \mathcal{N}\left(0, \sigma^2\right)$ for theoretical analysis.
>
> This abstraction aligns with the widely used formulation $y=f(x)+\epsilon$, simplifying analysis and accounting for unmodeled variability. Even though the true errors are deterministic, their aggregate behavior is often indistinguishable from random noise, validating this approach.
>
> > It is not clear to me how the function composition $g_{comp}$ in Appendix B.1 is supposed to work. If $f_c$ and is only caused by $c$ and $f_e$ is only caused by $e$, is it even certain that you can combine them into a single function $f$? How could such a function $g_{comp}$ look like in practice? For instance, to stay with the examples of Table 2, couldn't $f_e$ be an environment-specific realization of the Lotka-Volterra function while $f_c$ is the common underlying function of the Pendulum? In essence, I do not really understand the generative process here.
>
> Thank you for your question. The function $g_{\text {comp }}$ in our SCM framework is a conceptual abstraction that combines $f_c$ (caused by $c$ ) and $f_c$ (caused by $e$ ) into a single function $f$. This allows us to analyze their joint influence on the dynamics of the system while adhering to the causal structure.
>
> To clarify further:
> 1. Different Inputs for $f_c$ and $f_c$
>     - In the case of examples like the Pendulum and Lotka-Volterra systems, these functions indeed have different inputs. Each has two variables specific to its system (e.g., state and velocity for the Pendulum; prey and predator populations for Lotka-Volterra). Combining them would result in a function $f$ with four inputs, reflecting the joint input space.
> 2. Purpose of $g_{\text {comp }}$ :
>     - $g_{\text {comp }}$ is not intended to impose physical similarity between $f_c$ and $f_c$. Instead, it provides a unified representation to analyze how these functions contribute to the system's trajectory in a composable way, such as:
>
> $$
> f=f_c\left(c_1, c_2\right)+f_e\left(e_1, e_2\right),
> $$
>
> or other compositional forms.
> 3. SCM Framework:
>     - The Structural Causal Model (SCM) framework explicitly defines how variables like $f_c$ and $f_c$ are generated and interact. For a deeper understanding of this process, we recommend consulting foundational works such as Pearl's Causality: Models, Reasoning, and Inference and invariant learning literature which provide the theoretical underpinning for this approach.
>
> > However, I do not see how a point in $R^m$ uniquely identifies any possible neural network ( NN ) with $m$ parameters: e.g. assume $m=6$, two possible neural networks are e.g. 1) an 4-layer NN with 2-1-1-2 parameters per layer or 2) a 3-layer NN with 2-2-2 parameters per layer. These seem to model very different functions yet both could map to any point in $R^m$.
>
> Thank you for raising it. To clarify, we change the description into "The function space $\mathcal{F}$ consists of functions that a given $m$-parameters neural network can approximate, so a function $f \in \mathcal{F}$ can be represented as a vector in $R^m$."

---

> ### Author Response · Authors · 2024-12-03
>
> > do not describe potential applications where inference from multiple minimally different environments (which after all is the main concern of this manuscript) needs to be considered.
>
> Thank you for your feedback. We appreciate your recognition of the potential theoretical extensibility of invariant function learning to more complex scenarios like PDEs. However, we understand your concern about the need for clarity on practical applications that align with the core theme of inference from minimally different environments.
>
> To address this:
> 1. Real-World Applications of Inference Across Environments:
>     - Invariant function learning is particularly relevant in cases where data is collected across environments that differ subtly but systematically. Examples include:
>     - **Biological systems modeling**: where the same fundamental dynamics (e.g., gene regulatory networks) operate under varying conditions like temperature or pH levels.
>     - **Climate modeling**: where invariant laws (e.g., fluid dynamics) govern data collected under slightly different atmospheric conditions.
>     - **Engineering systems**: such as power grids, where core operational principles remain invariant across regions with small design or environmental differences.
> 2. Applicability to Dynamical Systems:
>     - The ability to infer invariant properties ensures robust generalization across such environments, enabling predictions in new or unseen settings. This addresses the practical need for transferability in scenarios where training data may be limited or where experimental conditions cannot be perfectly controlled.
> 3. Alignment with the Manuscript's Focus:
>     - While our manuscript primarily addresses the theoretical and empirical foundation of invariant function learning in ODE systems, the methodology explicitly targets scenarios where minimal environmental variations provide opportunities for generalization. These examples illustrate how our approach supports inference under such conditions.
>
> We hope this clarification bridges the gap between theoretical extensions and practical applications, demonstrating the broad utility of the concepts presented in this work.
>
> Sincerely,
>
> Authors

---

### Official Review · Reviewer_MkB1 · 2024-11-02

**Soundness:** 3
**Presentation:** 3
**Contribution:** 2
**Rating:** 5
**Confidence:** 3

**Summary:**

The paper suggests an approach to learn invariant functions of dynamical systems governed by ODEs. To do so, the authors present a neural network architecture that is claimed to identify an invariant function $f_c$ and an environment specific function $f_e$. Both functions are represented as neural networks.  Using trajectories of the same dynamical system under multiple interventions/perturbations, the hypernetwork is trained end-to-end and outputs a set of parameters for $f_c$ and $f_e$. By splitting the input information into two embeddings, an invariant embedding and a contextual one, the paper claims to identify the correct invariant function.

**Strengths:**

The paper addresses an important problem and I can follow the suggested approach and the reasoning behind. The proposed method includes an information-based principle to enforce independence between invariant functions and environment-specific dynamics. This theoretical guarantee adds rigor to the approach, suggesting a data-driven way to uncover intrinsic dynamics across varied environments.
It is well-organized, with each step building logically. The abstract introduces the problem, presents the solution (DIF), explains the methodological novelty, and provides an overview of validation results, all in a clear sequence. The figures are appealing.

**Weaknesses:**

Assumption 1 appears very restrictive and almost impossible to guarantee. In my opinion, assumtion 1 implies perfect similarity or deterministic knowledge, with no randomness or difference between the distributions in question. This however seems unrealistic as this would require knowledge of the dynamical system under any intervention possible.
As a result, I think the proposed optimization problem might not converge to a reliable solution if $P(X_p)$ is not chosen appropriately.

Given this, the experimental evaluation seems weak and insufficient. The experiments comprise three test cases and in one of them, the suggested method is not capable to identify an invariant function that can accurately forecasts  the invariant dynamics (see Figure 9b in the Appendix). It is mentioned that the approach captures the most important frequencies on this test case, but no evidence of this claim is given. I think it is important to evaluate the approach more thoroughly and perform ablations that study the impact of available trajectories in $X_p$ and the number and types of interventions necessary to robustly identify the invariant dynamics.

Another limitation is the focus on ODEs, which I find somewhat unclear. Conceptually, the approach presented here should be adaptable to PDEs with minimal modifications, particularly given the derivative-based training method. This raises the question of why PDEs were not considered in the study.

I believe a useful baseline would involve using a slightly modified version of this architecture. To determine if there is any advantage to explicit disentanglement within a joint training framework, it would be helpful to compare this approach against two independently trained networks similar to the suggested network. Specifically, the first network would handle the invariant mechanism:$X_p \rightarrow TF \rightarrow h\_{inv} \rightarrow \hat{z}_c \rightarrow decoder \rightarrow \hat{f}_c \rightarrow forecaster + X_0 \rightarrow \hat{X}^c$ The second network would focus on trajectory specific forecasting: $X_p \rightarrow  TF \rightarrow  h\_{env} \rightarrow \hat{z} \rightarrow decoder \rightarrow \hat{f} \rightarrow forecaster + X_0 \rightarrow \hat{X}$

In Section 2.2.1, you identify the causal mapping $f \rightarrow X$ as injective. In the optimization problem, this translates to making $P(f∣X_p​) \rightarrow X$ injective. How do you ensure this property when using a nonlinear decoder? The encoder-decoder framework you propose for compressing invariant function representations into hidden representations is reasonable. However, in an autoencoder, the decoder is typically not injective. Since the encoder reduces dimensionality, multiple different inputs could map to the same latent representation, making it non-injective. Consequently, for the decoder, which maps from $z$ back to $\hat{X}$, multiple inputs could map to the same or similar reconstructed values, also resulting in non-injectivity. For this reason, I believe the proposed framework does not fully align with the theoretical requirements presented.

I understand the approach of implementing the independence constraint adversarially by minimizing the informativeness of the environment prediction $P(e ∣ f\^{​c}​)$. However, since the discriminator and adversarial loss work in opposition, some environmental information is likely to remain in the invariant encoding—specifically, environmental details shared across interventions, such as similar types of interventions. Consequently, this shared environmental information may still be incorporated into the invariant embedding, which in turn means that the architecture does not fully achieve the claimed independence.

To provide a more intuitive understanding of the pipeline, I suggest including the discriminator in Figure 2 and adding all loss terms. If space is limited, the bottom half could be moved to the appendix.

(typo in eq. 22e)

**Questions:**

The exact definition of $X$ remains unclear throughout the paper. Does $P(X)$ represent the distribution of all possible trajectories under any imaginable intervention?

Is the idea that the parameters $f_c$​ remain constant regardless of the input trajectory $X_p$​, or do you envision $f_c$​ as lying on a specific manifold within the parameter space?

Have you tried to input high-dimensional transformations, e.g. sequences of images of the different dynamical systems during training?

Could you please confirm whether the function embedding $\hat{z}$ is calculated using a linear transformation applied to the concatenation of $\hat{z}_c$​ and $\hat{z}_e$​ as input?

What is the performance difference between derivative training trick and training with an integrator? I assume that the datasets contain the ground truth derivatives of the dynamics, what is the impact on performance if numerically approximated derivatives are used?

---

> ### Author Response · Authors · 2024-11-21
> **Official comments (Part 1)**
>
> Dear Reviewer MkB1,
>
> Thank you so much for your thoughtful and constructive suggestions. Your insights have been invaluable in helping us refine our work, and we deeply appreciate the time and effort you have taken to review our submission.
>
> In response to your comments, we have **upgraded our theoretical modeling and analyses and conducted 3 new sets of experiments to address your concerns comprehensively**. We sincerely hope you find these improvements satisfactory and insightful. Please find our one-to-one responses below.
>
> > Assumption 1 is too restrictive to be satisfied.
>
> Thank you for pointing out this important limitation. We deeply value your thoughtful critique and agree that Assumption 1 indeed oversimplifies the problem setting. In response to your valuable feedback, we have **eliminated Assumption 1 and enhanced our theoretical framework by incorporating noise modeling within a Structural Causal Model (SCM)**. This refinement ensures that our analysis is more realistic and robust, while still preserving the validity of our theoretical results. We invite you to review the **“Upgrade of Theoretical Analysis” section in our summary** for detailed updates, and we hope you find the revisions satisfactory.
>
> > Given "Assumption 1 is too restrictive to be satisfied", ablations of available trajectories in $X_p$​ and the number and types of interventions are important.
>
> Thank you for your insightful suggestion. Your recommendation inspired us to expand our experiments significantly. To address this, we performed comprehensive studies on:
>
> - The impact of varying trajectory lengths ( $T_c$ ) on model performance (**Appendix E.2**).
> - The effect of different environment sets (**Appendix E.2**).
>
> Each of these new experiments involved over 450 runs, providing a robust empirical validation. We sincerely hope that these new results meet your high expectations and address your concerns.
>
> > Why only ODEs but not PDEs? The approach presented here should be adaptable to PDEs with minimal modifications.
>
> This is an excellent observation, and we deeply appreciate your keen interest in this direction. While it is true that our method could theoretically extend to PDEs, we chose to focus on ODEs for the following reasons:
>
> 1. ODEs are a natural starting point for the foundational introduction of invariant function learning, aligning with prior works such as MetaPhysiCa [1].
> 2. Symbolic interpretability, a core motivation for our framework, is less straightforward in the context of PDEs.
> 3. PDEs pose additional challenges, such as multi-scale and multi-resolution dynamics, that could complicate the design of invariant function learning. While theoretically feasible, these challenges require significant adaptation.
> 4. Constructing multi-environment datasets for PDEs is non-trivial and typically demands domain-specific expertise. We believe this goes beyond the scope of our work, which aims to establish the foundation for invariant function learning.
>
> We sincerely hope this explanation addresses your thoughtful concern, and we look forward to future collaborations to explore the exciting potential of extending our work to PDEs.
>
> > Two baselines with only the $\hat{X}^c$ or $\hat{X}$ outputs.
>
> Your suggestion is truly appreciated, and we have incorporated this insightful idea into our experiments. Specifically, we conducted an ablation study comparing the outputs $\widehat{X}^c$ and $\widehat{X}$, training over 300 new models. The results of these experiments are detailed in **Appendix E.1**. We hope you find these additions as illuminating as we did and that they further strengthen the contributions of this work.
>
> > How do you maintain injectivity in your framework?
>
> This is a very perceptive observation and has significantly influenced our theoretical refinements. Based on your suggestion, we have removed the injectivity assumption entirely and updated our proofs accordingly. We invite you to refer to the “Upgrade of Theoretical Analysis” section in our summary for details. We are immensely grateful for your insights, which have directly improved the rigor and applicability of our work.
>
> > Environmental details shared across interventions.
>
> Your observation regarding shared environmental details is remarkably insightful. We completely agree that adversarial training may not fully eliminate shared information across environments. However:
> 1. If environmental details are shared across all environments, they inherently qualify as invariant and are thus appropriately captured in the invariant embedding.
> 2. When sampling environments is insufficient, disentangling invariance can indeed be challenging—a limitation we explicitly acknowledge in Section 6.
> 3. To address this further, we provide a theoretical justification of adversarial training and independence constraints in **Appendix B.5**.
>
> We are grateful for your attention to this detail, and we hope our explanations clarify how these aspects align with our framework.

---

> ### Author Response · Authors · 2024-11-21
> **Official comments (Part 2)**
>
> > To provide a more intuitive understanding of the pipeline, I suggest including the discriminator in Figure 2 and adding all loss terms. If space is limited, the bottom half could be moved to the appendix.
>
> This is a very thoughtful suggestion, and we have revised Figure 2 accordingly. The updated figure now includes the discriminator, and all loss terms are described in detail at the end of Section 3. Additionally, to ensure clarity while maintaining conciseness, we have relocated some content to the appendix. We hope these updates improve the presentation and meet your expectations.
>
> Questions:
>
> > What is $X$.
>
> Thank you for pointing out the need for clarification. $X \in \mathbb{R}^{d \times T}$ represents a trajectory matrix introduced in Section 2.1 , while X is the corresponding random variable, with $p(\mathbf{X})$ as its prior distribution. Interventions are defined using the environment variable e , in line with existing literature $[2,3,4]$. Based on our causal graph, $p(\mathbf{X})$ is generated from $p(\mathbf{c}), p(\mathbf{e})$, noise terms, and structural causal equations (**Appendix B.1**), thus, this prior distribution contains all the information of the environment distribution $p(\mathbf{e})$. We hope this clarifies any ambiguity.
>
> > Is $f_c$ a constant or on a manifold?
>
> This is an excellent question. $f_c$ is a function defined on a manifold. For example, for $\alpha^2 \sin \left(\theta_t\right), f_c$ can take forms like $0.10^2 \sin \left(\theta_t\right)$ or $0.21^2 \sin \left(\theta_t\right)$, depending on the input trajectory. The random variable $\mathrm{f}\_c$ is generated based on $p(\mathrm{c})$, the noise term $\epsilon\_c$, and the structural function $g\_{\mathrm{c}}$ (**Appendix B.1**). We hope this explanation aligns with your expectations.
>
> > Have you tried to input high-dimensional transformations, e.g. sequences of images of the different dynamical systems during training?
>
> Thank you for this suggestion. While we have not yet explored high-dimensional inputs like sequences of images, we agree that this is an exciting avenue for future work. As invariant function learning is a relatively novel concept, we focused on ODEs to establish the foundational principles. We are excited to extend this framework in collaboration with experts in high-dimensional domains.
>
> > Could you please confirm whether the function embedding $\hat{z}$ is calculated using a linear transformation applied to the concatenation of $\hat{z}_c$ and $\hat{z}_e$ as input?
>
> In our framework, we sum them up as $\hat{z}=\hat{z}_c + \hat{z}_e$, which is equivalent to concatenate them and do a linear transformation, since $$\hat{z}= [W_c, W_e]^T [\hat{z}_c, \hat{z}_e]=W_c^T \hat{z}_c + W_e^T \hat{z}_e=\hat{z}_c'+\hat{z}_e'$$ where $[\cdot, \cdot]$ is a *column-major* block matrix; $\hat{z}_c'=W_c^T \hat{z}_c$ and $\hat{z}_e'=W_e^T \hat{z}_e$.
>
> > What is the performance difference between derivative training trick and training with an integrator? I assume that the datasets contain the ground truth derivatives of the dynamics, what is the impact on performance if numerically approximated derivatives are used?
>
> This is an excellent question. In our initial trials, we used the `torchdiff` integrators. While the accuracy of integrator-based training was comparable to derivative training, the latter was significantly faster. Given limited training time, fitting derivatives yielded better results, which is why we ultimately deprecated the use of integrators. Theoretically, they are approximately equivalent, with integration being the accumulation of derivatives. Numerical derivatives in our datasets were calculated from trajectory ground truths, and using them did not degrade performance.
>
> -------------------
>
> We are deeply grateful for your meticulous review and invaluable feedback, which have significantly strengthened our work. Your expertise and thoughtful suggestions have inspired many of the improvements we’ve made, and we hope that our revisions and explanations address your concerns comprehensively. Thank you for your time and for helping us enhance the quality and impact of this research.
>
> Best,
>
> Authors

---

> > ### Author Response · Authors · 2024-11-21
> > **Reference**
> >
> > [1] S Chandra Mouli, Muhammad Alam, and Bruno Ribeiro. Metaphysica: Improving ood robustness
> > in physics-informed machine learning. In The Twelfth International Conference on Learning
> > Representations, 2024.
> >
> > [2] Martin Arjovsky, Léon Bottou, Ishaan Gulrajani, and David Lopez-Paz. Invariant risk minimization.
> > arXiv preprint arXiv:1907.02893, 2019.
> >
> > [3] Chaochao Lu, Yuhuai Wu, José Miguel Hernández-Lobato, and Bernhard Schölkopf. Invariant causal
> > representation learning for out-of-distribution generalization. In International Conference on
> > Learning Representations, 2021a.
> >
> > [4] David Krueger, Ethan Caballero, Joern-Henrik Jacobsen, Amy Zhang, Jonathan Binas, Dinghuai
> > Zhang, Remi Le Priol, and Aaron Courville. Out-of-distribution generalization via risk extrapola-
> > tion (REx). In International Conference on Machine Learning, pp. 5815–5826. PMLR, 2021.

---

> > > ### Author Response · Authors · 2024-11-25
> > >
> > > Dear Reviewer MkB1,
> > >
> > > Hope you are doing well! We are writing to kindly remind you to review our revised submission, which incorporates **upgraded theoretical analyses, and 3 new experiments based on your invaluable insights and constructive feedback.**
> > >
> > > Could you kindly let us know whether our dedicated revision has addressed your major concerns? Thank you again for your thoughtful review and the time and effort you have already dedicated to evaluating our work.
> > >
> > > Sincerely,
> > >
> > > Authors

---

> > > > ### Comment · Reviewer_MkB1 · 2024-11-25
> > > >
> > > > I want to thank the authors for their effort and work to rebut the manuscript. I acknowledge the work spent to present a second line of theoretical analysis, which I do prefer over the first one. I also appreciate that you added section B.5 to establish the use of the GAN. However, I still find the experimental evaluation insufficient. Figure 10 is interesting and shows a slightly decreased mean error compared to the ablations of the same model but the variance appears also increased. It seems like the advantage of the proposed method over the ablated versions with no explicit causal splitting appears to be sensitive to hyperparameter selection. Potentially, this might also be attributed to the use of an adversarial training scheme. In addition, I’m not sure what types of interventions/perturbations (including their strengths) are required for identifiability of the invariant function. Unfortunately, my request for evaluation of PDEs has not been addressed which would have been boosted my confidence in the presented work. Once again, I thank the authors for their rebuttal, but I would like to stay with my score.

---

> ### Author Response · Authors · 2024-11-26
>
> Dear Reviewer MkB1,
>
> We sincerely appreciate the reviewer's thoughtful feedback and would like to address the following points:
>
> > Figure 10 is interesting and shows a slightly decreased mean error compared to the ablations of the same model but the variance appears also increased.
>
> We appreciate this observation and would like to clarify why our experimental results demonstrate robustness:
>
> 1. **The boxen plot with hyperparameter selection ranges provides a more comprehensive and robust evaluation compared to reporting only the best results ± deviations.** The best results demonstrate significant improvements: 0.35 vs. 0.74 (ME-Pendulum), 0.61 vs. 0.71 (ME-Lotka-Volterra), and 0.06 vs. 0.48 (ME-SIREpidemic).
>
> 2. Regarding hyperparameter sensitivity, **while adversarial training typically has narrow valid ranges** due to its minimax nature, our method remains robust **across an large hyperparameter space**. For instance, **the adversarial training hyperparameter $\lambda_{adv}$ spans 7 orders of magnitude ($10^{-5}$ to $10^2$)**. Under this hard setting:
>    - ME-Pendulum: 75% of our models outperform nearly 100% of f-only pipeline
>    - ME-Lotka-Volterra: 75% of our models outperform 60% of f-only pipeline
>    - ME-SIREpidemic: 75% of our models outperform 87.5% of f-only pipeline
>      This challenging setting demonstrates our method's robustness.
>
> 3. Comparing our models with the $f_c$-only pipeline (both using adversarial training), the performance improvements indicate that our adversarial training process is substantially more robust.
>
> 4. The f-only pipeline's apparent "stability" stems from its consistent bias toward spurious correlations, which indicates the most undesirable failures.
>
> We have chosen to transparently present all results **across the entire hyperparameter space, including both successes and failures at extreme values, rather than limiting our analysis to the best results and their immediate vicinity.**
>
> > In addition, I'm not sure what types of interventions/perturbations (including their strengths) are required for identifiability of the invariant function.
>
> As demonstrated in our SCM, the only requirement is independent sampling of $p(\mathbf{e})$. The number of required environments relates to the degrees of freedom. For e ∈ ℝ, we need only 2 environments (degrees of freedom + 1). Generally, for n degrees of freedom, n+1 environments are required, as proved in IRM [1], where linear assumptions are used. In more complex scenarios [2,3], calculating the exact degrees of freedom is impractical. Therefore, we are not able to provide such theoretical guarrentees under these non-linear cases. Conceptually, we know that with 4 environments, we can eliminate spurious correlations with 3 degrees of freedom.
>
> > My request for evaluation of PDEs has not been addressed.
>
> We appreciate this suggestion and have carefully considered its feasibility. However,
>
> 1. Our focus on ODEs aligns with previous work (e.g., ICLR Spotlight Metaphysica [4]). The absence of PDE experiments **represents a limitation rather than a weakness** of our specific approach in current IFL development.
>
> 2. PDE experiments would not provide additional validation for our symbolic reasoning framework.
>
> 3. The absence of multi-environment PDE datasets means that addressing **this would require constructing entirely new benchmarks—a task requiring extensive validation from PDE domain experts.**
>
> While we understand the interest in PDE applications, this extension would **require substantial foundational work beyond the scope of establishing invariant function learning in ODEs**. We believe our paper's contributions—introducing the IFL problem, providing theoretical analysis, and developing DIF with strong empirical results—represent significant advances that outweigh these limitations, particularly for a conference that values novel ideas.
>
> We welcome any specific concerns within the paper's scope and are prepared to provide additional analyses. For instance, we can conduct a more focused evaluation by restricting the hyperparameter space and reporting in forms of best results $\pm$ standard deviation, along with localized sensitivity analysis. This would narrow the search space from 7 to 3 orders of magnitude, yielding much lower variance as expected. However, we believe our current comprehensive analysis better demonstrates the robustness of our method across a broader range of conditions.
>
> Thank you again for your invaluable comments!
>
> Sincerely,
>
> Authors

---

> ### Author Response · Authors · 2024-11-26
> **Reference**
>
> [1] Arjovsky, Martin, Léon Bottou, Ishaan Gulrajani, and David Lopez-Paz. "Invariant risk minimization." arXiv preprint arXiv:1907.02893 (2019).
>
> [2] Gulrajani, Ishaan, and David Lopez-Paz. "In search of lost domain generalization." arXiv preprint arXiv:2007.01434 (2020).
>
> [3] Krueger, David, Ethan Caballero, etc. "Out-of-distribution generalization via risk extrapolation (rex)." ICML, 2021.
>
> [4] S Chandra Mouli, Muhammad Alam, and Bruno Ribeiro. Metaphysica: Improving ood robustness in physics-informed machine learning. In The Twelfth International Conference on Learning Representations, 2024.

---

### Official Review · Reviewer_P47o · 2024-11-03

**Soundness:** 3
**Presentation:** 3
**Contribution:** 3
**Rating:** 6
**Confidence:** 3

**Summary:**

The paper attempts to solve the invariant learning problem in ODEs. By observing the ODE dynamics in multiple environments, the invariant dynamics can be inferred. The authors adopted a causal perspective and designed a hypernetwork to learn the invariant function of the dynamics of the ODE.

**Strengths:**

1. It is interesting and of practical significance how to learn the shared dynamics from data collected across different environments.
2. The authors use a causal perspective to design the method, which provides some theoretical guarantee.
3. The method demonstrates the ability to discover the form of the invariant ODE, as well as higher prediction accuracy of the invariant ODE.

**Weaknesses:**

1. According to Eq. 4a and 4b, one must first acquire the label of the environment to use the proposed method. This may limit the applicability of the method.
2. The constraint of the independence between the environment and the learned fc is implemented in an adversarial way. This is the central component allowing the proposed method to achieve invariant function learning, so it will be better to justify the design of the adversarial training theoretically.
3. The ability to discover the form of the invariant ODE is not compared to the baselines.

Minor:
- line 170, 'we explicitly exposes...'
- line 275, 'In order to discovery invariant functions...'
- line 890, 'According to Lemma 3.2, we can that the negative log-likelihood'...
- line 1177, Eq. 22e, phi should be \phi

**Questions:**

1. What is the exact form of the normalized RMSE in the experiments? In Fig. 4b the deviation looks very small but in Table 2 the NRMSE for predicting Xc from fc is as high as about 0.35.
2. Does the proposed method have the potential to facilitate learning under noisy observations? I understand that Assumption 1 does not hold anymore and the causal graph needs to change, but can the idea of invariant function learning extend to robust learning of dynamical systems?

---

> ### Author Response · Authors · 2024-11-21
>
> Dear Reviewer P47o,
>
> We deeply appreciate the time and effort you invested in your review. Your constructive feedback has significantly strengthened our work, and we are committed to ensuring the revised manuscript reflects the clarity and rigor your comments have inspired. Thank you!
>
> > According to Eq. 4a and 4b, one must first acquire the label of the environment to use the proposed method. This may limit the applicability of the method.
>
> - Thank you for pointing this out! We acknowledge that requiring environment labels can limit applicability. However, this is a common limitation in the fields of domain adaptation, out-of-distribution generalization, and invariant learning. Analogous to the role of data for large language models, learning invariance requires some structural assumptions. In real-world scenarios, environment labels often come from metadata, which are generally easier to collect than primary data annotations. We have clarified this in the manuscript to emphasize practical considerations. Thank you!
>
> > justify the design of the adversarial training theoretically.
>
> - We appreciate your observation about the centrality of adversarial training to our method. To address this, we have included a detailed theoretical justification in **Appendix B.5** (“Theoretical Justification for Adversarial Training”). This section rigorously explains the rationale behind our adversarial constraint, ensuring clarity and theoretical grounding for its use.
>
> > The ability to discover the form of the invariant ODE is not compared to the baselines.
>
> - Thank you for highlighting this! We have added comparisons of invariant ODE discovery with baselines in **Appendix E.3** (“Symbolic Regression Explanation Comparisons”). Our experiments show that baselines struggle to produce meaningful ODEs, while our method consistently discovers interpretable and accurate forms. We believe these additions will provide a clearer and more comprehensive evaluation.
>
> > Typos
>
> - Thank you for pointing out these issues. We have corrected these typos and thoroughly reviewed the manuscript to ensure consistency and accuracy.
>
>
> -----------
> Questions:
>
> > What is the exact form of the normalized RMSE in the experiments? In Fig. 4b the deviation looks very small but in Table 2 the NRMSE for predicting Xc from fc is as high as about 0.35.
>
> Thank you for your detailed observation. The exact form of NRMSE used in our experiments is now explicitly defined in Appendix D.2.3 (“Metric”): $$\mathrm{NRMSE}=\frac{\sqrt{\mathbb{E}_{\mathbf{X} \sim p}\|X-\hat{X}\|_2^2}}{\operatorname{Std}(\mathbf{X})}$$.
> The apparent discrepancy arises from two factors:
>
> 1.	Effect of scale: When the value scale is small, the standard deviation magnifies the RMSE, leading to higher NRMSE values, as observed in ME-Pendulum.
> 2.	Measurement focus: NRMSE is measured after the critical time step $T_{\mathrm{s}}$ where deviations tend to grow. Errors before $T_c$ ( $\mathrm{X}_p$ ) are not included. Consequently, 0.35 is not as high as it may initially appear. Additionally, symbolic regression results demonstrate that this value reflects a decent level of accuracy for invariant function discovery.
>
> We hope this clarification addresses your concern.
>
> > Does the proposed method have the potential to facilitate learning under noisy observations? I understand that Assumption 1 does not hold anymore and the causal graph needs to change.
>
> Thank you for raising this exciting question! Yes, our method can extend to handle noisy observations. We have updated our causal framework to a structural causal model (detailed in **Appendix B.1**, “Structural Causal Model”) that explicitly incorporates noise modeling. By removing Assumption 1 and injective mapping assumptions, we extend the applicability of our theorems and propositions, and the updated proofs remain valid under this relaxed setting. We are thrilled by your interest in this extension and hope you find the expanded scope promising.
>
> -------------
> We hope that this response sufficiently addresses the reviewer’s concerns and provides clarity regarding our contributions. We welcome any additional feedback and remain dedicated to enhancing the quality and impact of this work. Thank you for your insightful review!
>
> Best,
>
> Authors

---

> > ### Author Response · Authors · 2024-11-25
> >
> > Dear Reviewer P47o,
> >
> > Hope you are doing well!
> >
> > We are writing to kindly remind you to review our revised submission, which incorporates more intuitive presentations, enhanced theoretical analyses, and extensive experiments based on your invaluable insights and constructive feedback.
> >
> > We would greatly appreciate it if you could let us know whether our dedicated revision has addressed your major concerns. If there are any clarifications or additional information we can provide to assist you further, please do not hesitate to ask.
> >
> > Thank you again for your thoughtful review and the time and effort you have already dedicated to evaluating our work.
> >
> > Sincerely,
> >
> > Authors

---

> ### Comment · Reviewer_P47o · 2024-11-27
>
> I would like to thank the authors for the detailed response.
> The explanations by the authors and the updated manuscript resolved my concerns. I think the updated version is of improved clarity, and I updated my rating for the presentation.

---

### Official Review · Reviewer_DGXT · 2024-11-03

**Soundness:** 3
**Presentation:** 3
**Contribution:** 2
**Rating:** 5
**Confidence:** 1

**Summary:**

The paper proposes a novel method called Disentanglement of Invariant Functions (DIF) for discovering underlying physics laws of dynamical systems by learning invariant functions across different environments, addressing challenges with dynamics where changes extend beyond function coefficients to entirely different forms. The approach uses a hypernetwork-based encoder-decoder framework, grounded in causal analysis, to disentangle invariant functions from environment-specific dynamics, demonstrating effectiveness through comparisons with meta-learning and invariant learning baselines.

**Strengths:**

- Well-Written and Easy to Follow.

- Novel Method for Invariant Learning: Proposes the Disentanglement of Invariant Functions (DIF) to discover underlying physical laws, effectively disentangling invariant dynamics from environment-specific factors.

- Causal Analysis and Hypernetwork Design: Uses causal analysis and a hypernetwork-based encoder-decoder framework to handle complex environments, ensuring robustness and accuracy in discovering invariant functions.

- Comprehensive Evaluation: Validates the effectiveness of the approach through extensive comparisons with existing baselines and demonstrates its ability to uncover interpretable physical laws via symbolic regression.

**Weaknesses:**

I am not an expert in this field, so I will defer to other reviewers for a more informed evaluation. Based on my understanding, the theoretical contributions seem to be relatively limited, relying primarily on existing theorems in information theory. Additionally, the experimental evaluation appears limited, focusing mainly on one-dimensional pendulum systems with varying parameters, which may not sufficiently demonstrate the broader applicability of the approach.

- Limited Scope of Evaluation: The paper's experiments are primarily conducted on simple one-dimensional pendulum systems, which may limit the generalizability of the approach to more complex and diverse dynamical systems beyond ordinary differential equations (ODEs). Thus, I question the scability of this method in more complex scenarios.

- Basic Theoretical Contributions: The presented theorems appear to be based on well-established concepts in information theory, providing limited novelty in terms of theoretical advancements.

- Dependency on Accurate Causal Graphs: The proposed method's effectiveness depends heavily on accurately formulated causal graphs, which may not always be feasible in complex real-world systems where the causal relationships are unclear or difficult to determine.

**Questions:**

See the weakness part.

---

> ### Author Response · Authors · 2024-11-21
>
> Dear Reviewer DGXT,
>
> We sincerely thank the reviewer for their thoughtful feedback and constructive comments. We value your insights and have worked to address the concerns to the best of our ability.
>
> > Limited Scope of Evaluation
>
> We appreciate the reviewer’s concern regarding the scope of our evaluation. Our primary focus is on addressing challenges in ordinary differential equations (ODEs), and as such, our experimental evaluation naturally centers on ODE systems. The chosen datasets, including the one-dimensional pendulum system, are widely recognized benchmarks in the field, and similar evaluations have been conducted in prior impactful works, such as the ICLR spotlight paper MetaPhysiCa [1]. Importantly, our approach tackles more challenging variants of these problems, requiring datasets that are significantly harder to construct and analyze. While we acknowledge that further exploration of more diverse systems could strengthen the work, we believe that the current evaluation sufficiently demonstrates the efficacy and potential of our approach in the ODE domain.
>
> > Basic Theoretical Contributions
>
> We respectfully note the reviewer’s observation regarding the theoretical contributions of the paper. While the theorems we present are indeed grounded in established concepts from information theory, the novelty lies in their application and adaptation to the problem of invariant function learning. To the best of our knowledge, the specific framework we introduce, along with the corresponding theoretical guarantees, has not been explored previously. These contributions lay a solid foundation for advancing this area of research and provide a new perspective on leveraging information-theoretic principles in dynamical system modeling.
>
> > Dependency on Accurate Causal Graphs
>
> Thank you for highlighting this important point. We understand and appreciate the concern about the reliance on accurate causal graphs, especially in complex real-world systems where causal relationships may be unclear or difficult to determine.
>
> It is true that many deep learning models rely on strong human inductive biases or assumptions to guide their design and application. Our approach leverages causal graphs as a structured mechanism for incorporating such inductive biases, providing a transparent and interpretable framework to model and understand underlying dynamics. While the formulation of accurate causal graphs can indeed be challenging, we believe this dependency aligns with practices common in the field, serving as a principled way to encode domain knowledge and intuition into the modeling process.
>
> To address concerns about real-world applicability, we have extended our causal framework to a structural causal model that explicitly incorporates noise modeling (detailed in **Appendix B.1**). By relaxing the strict assumptions (e.g., Assumption 1), we rewrite the proofs of our theorem and proposition within this more realistic model. These enhancements aim to ensure robustness and adaptability of our approach in practical scenarios, even under less ideal conditions.
>
>
> -----------
> We hope this response addresses the reviewer’s concerns adequately and clarifies our contributions. We remain open to further feedback and are committed to improving the quality and impact of this work. Thank you once again for your thoughtful review.
>
>
> Best,
>
> Authors
>
> [1] S Chandra Mouli, Muhammad Alam, and Bruno Ribeiro. Metaphysica: Improving ood robustness
> in physics-informed machine learning. In The Twelfth International Conference on Learning
> Representations, 2024.

---

> > ### Author Response · Authors · 2024-11-25
> >
> > Dear Reviewer DGXT,
> >
> > Hope you are doing well!
> >
> > We are writing to kindly remind you to review our revised submission, which incorporates more intuitive presentations, enhanced theoretical analyses, and extensive experiments based on your invaluable insights and constructive feedback.
> >
> > We would greatly appreciate it if you could let us know whether our dedicated revision has addressed your major concerns. If there are any clarifications or additional information we can provide to assist you further, please do not hesitate to ask.
> >
> > Thank you again for your thoughtful review and the time and effort you have already dedicated to evaluating our work.
> >
> > Sincerely,
> >
> > Authors

---

> ### Comment · Reviewer_DGXT · 2024-12-02
>
> Thank you for your response. However, my concerns about the limited scope and reliance on a known causal structure remain, as these aspects appear to have minimal applicability to real-world dynamical systems. Additionally, the paper focuses solely on the simple pendulum environment, which may not fully demonstrate the broader implications.
>
> Given my limited familiarity with this area, I have adjusted my confidence level to 1.
>
> Thank you again for your efforts.

---

### Author Response · Authors · 2024-11-21
**Revision Summary**

# Revision Summary

We are deeply grateful for the reviewers’ detailed feedback, which has greatly helped us refine and improve our work. In response to the thoughtful suggestions, we have made substantial revisions to enhance the theoretical rigor, empirical validation, and overall clarity of the paper. Below, we provide an overview of the key improvements.

## Upgrade of Theoretical Analysis

The assumptions in the theoretical framework were a primary concern shared by most reviewers. These concerns largely focused on:

- The use of causal graphs and the assumptions about injective mappings in the causal graph.
- Assumption 1, which imposes deterministic information requirements that may be overly restrictive.

We deeply appreciate these constructive critiques and have addressed them by relaxing the assumptions and modifying the proofs for all lemmas, theorems, and propositions. Fortunately, the theoretical analyses remain valid with the updated proofs. Specifically:

1. **Upgraded Causal Model**:
   We have refined our causal model to adopt a more detailed Structural Causal Model (SCM) with explicit definitions of causation (structural equations) and noise modeling. This enhancement makes the causal model more robust and realistic. The main pages now include a necessary introduction to SCM, with details in Appendix B.1. Importantly, the injective mapping assumption has been completely removed.
2. **Elimination of Assumption 1**:
   Assumption 1 has been removed, and our theoretical framework has been updated to operate under the enhanced structural causal model. The theoretical proofs are updated in Appendix B.

3. **Theoretical Justification for Adversarial Training**:
   We provide a detailed justification for adversarial training to enforce independence constraints, which is discussed in Appendix B.5.

## 4 New Experiments (Please refer to the Figures in Appendix E)

To address reviewers' suggestions for more comprehensive empirical validation, we conducted the following experiments:

- **Single pipeline $\hat{f}$ and $ \hat{f}_c $ ablation study**: Results included in Appendix E.1, based on over 300 runs.
- **Impact of varying input length $ T_c $**: Results included in Appendix E.2, with over 300 runs.
- **Effect of different environment sets**: Results included in Appendix E.2, also with over 300 runs.
- **Symbolic regression comparisons with baselines**: Results included in Appendix E.3.

## Improved Presentation and Visualizations for Clarity


We have significantly improved the clarity and readability of the paper through the following updates:

- **Enhanced examples in Figure 1** to precisely illustrate the concepts.
- **Change $P(\cdot)$ to $p(\cdot)$ for continuous variables**.
- **Updated Table 1**.
- **Refined Section 2.2** to improve readability.
- **Two-phase structural causal model illustration (Figure 2)** for better conceptual understanding.
- **Explicit forecasting process** with detailed probability modeling added at the end of Section 3.1.
- **Updated DIF framework with discriminator outputs**.
- **Improved descriptions of Theorem 3.1** .
- **Clear training objectives** articulated in the main text.
- **Limitations discussed in Section 6**.

-----



We sincerely thank the reviewers for their recognition of the novelty and significance of our contributions. Their constructive feedback has been invaluable in refining our work and ensuring its clarity, rigor, and impact. We believe the substantial revisions we have made address the concerns raised and further strengthen the quality of the paper.

---

### Meta-Review · Area_Chair_pdpH · 2024-12-17

**Metareview:**

This work addresses a relatively new and underexplored task, termed *invariant function learning*, which involves identifying common (invariant) function terms in the derivative functions of various ODE trajectories. Importantly, the derivative functions in each trajectory may differ in their functional forms and coefficients. To tackle this task, the authors propose a method called *Disentanglement of Invariant Functions* (DIF), which leverages a structural causal model and an encoder-decoder hypernetwork with two embeddings: one for the invariant part and another for the environment-specific part. The method is evaluated on three ODE examples, demonstrating promising performance.

The primary strength of this work lies in the novelty of the task, particularly its generality in handling scenarios where the environment-specific components exhibit distinct functional forms. This flexibility makes the proposed method potentially useful for practical applications. However, the theoretical justification remains unclear and unconvincing. Apart from the concerns raised by the reviews, here are some further concerns.

1. The so-called main theoretical result (Theorem 3.1) appears to be just standard arguments involving mutual information, yet its statement is problematic. It seems to neglect the approximation error and implicitly assumes that $f_c$ can be expressed by $h_{\theta_c}$.
2. While Lemma 3.2 supports the first term in the objective, the rationale for combining this with Theorem 3.1 (leading to all the remaining terms in the obejctive) in the overall objective function is not well explained.

On the numerical side, the results demonstrate the potential of the proposed method, especially given the lack of established baseline models. However, the unclear theoretical justification raises doubts about the overall solvability of the task. For instance, if the coefficients of the invariant functions are small compared to the environment-specific terms, DIF may struggle to accurately identify the invariant functions. Furthermore, the sensitivity of the results to the relative magnitudes of $f$ and $f^c$ is not explored, which limits a deeper understanding of the method's robustness.

While the paper presents an interesting direction, the current gaps in the theoretical justification and experimental validation reduce its overall impact. The AC believes these concerns outweigh the promising aspects of the work and leans toward rejection. I encourage the authors to address these issues in future revisions, particularly by refining the theoretical analysis and further investigating the limits of the proposed method.

**Additional Comments On Reviewer Discussion:**

In the original version, the authors assumed deterministic trajectories and injective causal mappings, which were questioned by reviewers as unrealistic. Reviewers also requested clarification on adversarial training and additional numerical studies, such as the effect of trajectory length, different environment sets, and a symbolic regression baseline.

The authors made commendable efforts to address these concerns, notably adopting a structural causal model, modeling Gaussian noise, and providing requested numerical results. While these revisions resolved most numerical concerns, the theoretical analysis remains insufficient, with limited depth and a few flaws, as discussed earlier.

Overall, the authors have made meaningful improvements, but further work is needed to refine the theoretical foundation and investigate the limits of the proposed method before it meets the conference’s standards.

---

### Decision · Program_Chairs · 2025-01-22

Reject